# BIT-LLM: Brain Instruction Tuned LLM with Persistent Cross-Attention for fMRI-to-Text Decoding

Sunghwan Lee [1]   Jihun Kim [1]   Chae Lynn Kim [1]   Ji Yun Park [1]   Jong-Hwan Lee [1]

## Abstract

Decoding fMRI into natural language is challenging because strong, pre-trained language priors can dominate autoregressive generation, obscuring whether a model truly utilizes neural evidence. We introduce Brain Instruction Tuned LLM (BIT-LLM), which exposes fMRI-derived tokens as a persistent key–value memory through interleaved cross-attention adapters, enabling repeated neural access throughout decoding. BIT-LLM is trained with a three-stage pipeline: (i) multimodal contrastive learning to obtain semantically aligned fMRI representations, (ii) supervised fine-tuning to learn the brain-LLM interface while freezing the encoder and backbone LLM, and (iii) reward-based finetuning to optimize sequence-level caption quality directly. On the NSD subject-heldout (S1–7 train, S8 test), BIT-LLM yields substantially improved captioning quality over prior baselines under greedy decoding. In addition to standard captioning metrics, we perform several complementary evaluations to assess the robustness of brain–language grounding. Specifically, we conduct perturbation-based sanity checks by zeroing fMRI inputs or shuffling voxel values, and examine whether internal representations and generated outputs change accordingly. BIT-LLM exhibits clear sensitivity to these perturbations, indicating effective utilization of voxel values and their spatial correspondence.

## 1. Introduction

Decoding human brain activity into natural language from functional MRI (fMRI) is a central challenge in Neuro-AI, with potential applications to non-invasive brain-computer interfaces and computational accounts of visual-semantic representations. A key practical requirement is subject-agnostic decoding: training a single model that generalizes across individuals.

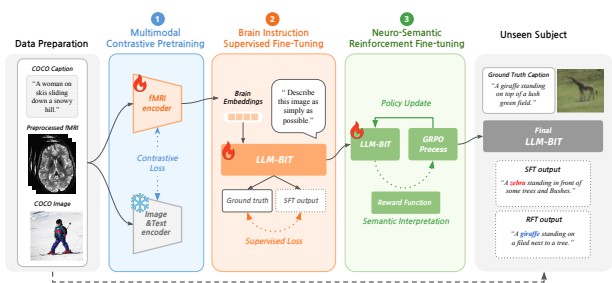

*Figure 1.* **Overview of the BIT-LLM framework.** The pipeline consists of three training stages: (1) Multi-modal Contrastive Pretraining aligns fMRI signals with image and text embeddings; (2) Brain Instruction SFT trains the brain-LLM interface using a persistent cross-attention mechanism; and (3) Neuro-Semantic RFT optimizes the model with reinforcement learning using sequence-level rewards. Finally, the model generates captions for unseen subjects (Subj08) solely from fMRI signals.

Recent fMRI-to-text systems typically project neural features into the embedding space of a pretrained LLM and inject them once as continuous prefix tokens (Li & Liang, 2021; Qiu et al., 2025; Xia et al., 2024). However, one-shot conditioning can weaken during autoregressive generation: language models may under-utilize information that appears early in the context and fall back on strong linguistic priors (Khandelwal et al., 2018; Liu et al., 2024; Li et al., 2024). As a result, high captioning scores alone do not necessarily imply that generations are grounded in voxel values rather than prior-driven shortcuts.

We introduce **BIT-LLM**, a subject-agnostic fMRI-to-text framework that treats fMRI as a first-class modality via persistent brain cross-attention. Instead of conditioning only at the input, BIT-LLM maintains fMRI-derived tokens as an external key-value memory and interleaves lightweight cross-attention adapters across multiple depths, enabling intermediate text states to repeatedly query neural evidence during decoding, in the spirit of modern multimodal LLM interfaces (Dai et al., 2024).

[1]Department of Brain and Cognitive Engineering, Korea University, Seoul, Republic of Korea. Correspondence to: Jong-Hwan Lee <jonghwan_lee@korea.ac.kr>.

*Proceedings of the $43^{rd}$ International Conference on Machine Learning*, Seoul, South Korea. PMLR 306, 2026. Copyright 2026 by the author(s).

As shown in Figure 1, BIT-LLM is trained with a three-stage pipeline: **(1) Multimodal contrastive pre-training** aligns fMRI with paired images and text in a CLIP-style space (Radford et al., 2021); **(2) Supervised Fine-Tuning (SFT)** learns the brain-LLM interface for caption generation while freezing the backbone to preserve linguistic competence; and **(3) Neuro-Semantic Reinforcement Fine-Tuning (RFT)** optimizes sequence-level rewards for caption quality and semantic grounding (e.g., CIDEr (Vedantam et al., 2015) and BERTScore (Zhang et al., 2020)) using GRPO (Shao et al., 2024).

On the Natural Scenes Dataset (NSD) (Allen et al., 2022) under the standard subject-heldout protocol (S1–7 train, S8 test), BIT-LLM improves captioning quality over prior systems and exhibits stronger neural reliance under voxel perturbation diagnostics (Appendix B).

**Contributions** We make three contributions: **(i) Persistent neural access:** an interleaved brain cross-attention interface that mitigates one-shot prefix conditioning decay in long-form decoding; **(ii) Grounded representations and diagnostics:** contrastive pre-training yields value-dependent brain embeddings, and perturbation tests reveal end-to-end reliance on voxel values and value–location correspondence (Appendix B); **(iii) Strong held-out subject decoding:** improved NSD subject-held-out captioning over prior baselines under the same protocol.

**Conflict of Interest Disclosure.** The authors declare no financial conflicts of interest related to this work.

## 2. Related Work

### 2.1. From fMRI to Semantic Representations

Non-invasive brain decoding has progressed from identifying viewed stimuli and reconstructing visual content to recovering higher-level semantic representations and natural language from fMRI (Kay et al., 2008; Naselaris et al., 2009; Nishimoto et al., 2011; Tang et al., 2023; Ye et al., 2025). A common strategy is to map fMRI patterns into a shared multi-modal embedding space and then leverage retrieval or strong generative priors (Scotti et al., 2023; Ma et al., 2025). For example, CLIP-pivot pipelines align fMRI to CLIP embeddings and employ external priors for reconstruction or open-vocabulary decoding (Scotti et al., 2023; Ma et al., 2025). These results motivate treating fMRI as a semantic signal, but they often rely on task-specific modular pipelines; in this work we take a first step toward unification by focusing on the fMRI-to-caption setting under a held-out subject protocol on NSD (Allen et al., 2022).

### 2.2. LLM Interfaces for Brain-Conditioned Generation

Recent brain-to-text systems couple subject-agnostic fMRI encoders with pretrained LLMs for open-vocabulary generation, commonly via a one-shot connector that injects brain features only at the input (Qiu et al., 2025). In multi-modal LLMs, two connector patterns are prevalent: (i) prompt/prefix-style projections that are consumed once (Tsimpoukelli et al., 2021; Li et al., 2023; Liu et al., 2023a), and (ii) interleaved (often gated) cross-attention adapters that allow intermediate text states to repeatedly retrieve non-text evidence from an external memory (Alayrac et al., 2022). Unified frameworks such as OneLLM further generalize this connector view across modalities (Han et al., 2024). Our setting follows the second pattern: BIT-LLM treats fMRI-derived tokens as an external key-value memory and interleaves brain cross-attention adapters to provide persistent neural access throughout decoding, addressing the failure mode where one-shot conditioning weakens over long autoregressive generation (Table 4).

### 2.3. Sequence-Level Post-Training for Grounding

Post-training objectives are widely used to align generators beyond token-level likelihood, including RLHF-style reward optimization and preference-based methods (Ouyang et al., 2022; Rafailov et al., 2023). GRPO is a PPO-style objective that uses group-relative advantages for efficient updates without an explicit value network (Shao et al., 2024). Such ideas have been extended to multimodal alignment to improve factuality and grounding with respect to non-text evidence (Yu et al., 2024; Sun et al., 2023). Motivated by this line, our Neuro-Semantic RFT applies GRPO with rewards tailored to brain-conditioned caption quality and reference-based semantic alignment. To reduce post-training cost while preserving (and empirically improving) performance, we implement the RFT updates via Low-Rank Adaptation (LoRA) (Hu et al., 2022), and observe that enabling LoRA yields better post-training gains than training without LoRA under an otherwise identical recipe (Appendix E.11).

## 3. Method

### 3.1. Model Overview

Our goal is to decode natural-language descriptions directly from fMRI by treating brain activity as a first-class modality within a large language model (LLM). BIT-LLM couples (i) a subject-agnostic fMRI encoder that maps each trial to a compact set of brain tokens, with (ii) a decoder-only LLM augmented with brain cross-attention adapters. Figure 2 summarizes the architecture.

Concretely, we use a neuroscience-informed fMRI encoder (Qiu et al., 2025) and a Llama 3.2-3B-Instruct backbone (Dubey et al., 2024). The encoder produces brain tokens

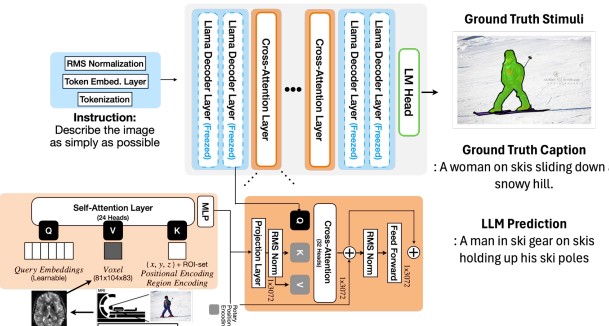

*Figure 2.* BIT-LLM model architecture. An fMRI encoder converts cortical responses into a sequence of brain tokens, which are projected and provided as a persistent key–value memory to a decoder-only LLM via interleaved brain cross-attention adapters.

that are projected to the LLM hidden size and exposed to the LLM through cross-attention. During generation, text hidden states act as queries while brain tokens provide keys and values, enabling the decoder to repeatedly retrieve neural evidence at selected depths of the transformer stack. Our cross-attention design follows gated, adapter-style multimodal decoders that condition a largely frozen language backbone by inserting cross-attention between pretrained transformer blocks (Alayrac et al., 2022).

### 3.2. fMRI Encoder and Brain Tokens

Given a preprocessed fMRI sample $x_{s,i}$ from subject $s$, we extract voxel-wise responses within a cortical mask and encode them with a subject-agnostic, neuroscience-informed attention encoder (Qiu et al., 2025). The encoder produces a sequence of brain tokens $\mathbf{B}_{s,i} \in \mathbb{R}^{T_b \times d_e}$ that summarizes stimulus-evoked activity while accommodating variable voxel layouts across subjects. We project tokens to the LLM hidden dimension and use them as the key–value memory for brain cross-attention (Section 3.3).

### 3.3. Brain–LLM Integration via Cross-Attention

**Prefix conditioning as a baseline interface** A common strategy for conditioning LLMs on non-text modalities is to encode the auxiliary input into a small set of latent vectors, project them into the LLM hidden space, and prepend them as prefix tokens (Qiu et al., 2025; Han et al., 2024; Li et al., 2025; Shen et al., 2025). While simple and parameter-efficient, prefix conditioning is effectively one-shot: the conditioning signal is injected only at the input and must be preserved implicitly through the transformer stack. We use prefix conditioning as a controlled baseline in experiments.

**Our design: persistent neural access via interleaved cross-attention** We instead treat fMRI-derived tokens as a persistent external key-value memory and expose them through interleaved brain cross-attention adapters, which

allow intermediate text states to repeatedly query neural evidence during autoregressive decoding (Alayrac et al., 2022; Dai et al., 2024). Concretely, given brain tokens $\mathbf{B}$ from the fMRI encoder, we project $\mathbf{B}$ to the LLM hidden dimension and use them as cross-attention keys/values, while language hidden states act as queries. This keeps modality features separate from the text token stream and avoids inflating the effective text sequence length (in contrast to concatenation-based conditioning) (Dai et al., 2024).

**Insertion schedule (4+1 template; early instantiation)** A key design knob is insertion density: denser interleaving typically strengthens conditioning but increases compute and activation/memory cost, motivating sparse schedules in large-scale multimodal settings (Alayrac et al., 2022). For the 28-layer Llama 3.2-3B backbone (Dubey et al., 2024), we adopt a sparse **4+1** template and instantiate the first four cross-attention adapters in the early portion of the stack. Operationally, this yields an early-biased, low-cost interface that introduces neural conditioning without perturbing the backbone's late-layer language modeling. Table 1 shows that, under a fixed adapter budget (four adapters), depth placement has only a minor effect on captioning quality (early- vs. late-biased configurations are nearly tied). Notably, in our brain-conditioned setting, increasing the number of inserted adapters does not yield monotonic gains, suggesting diminishing returns and the need for a budgeted insertion strategy. Consequently, we adopt four early-biased adapters as a simple efficiency-accuracy default in all main experiments, and treat insertion density as a key design trade-off.

**Why cross-attention for fMRI** Cross-attention is a standard connector for conditioning a language model on external modality representations through dedicated adapter layers, enabling controlled, repeated retrieval during generation (Alayrac et al., 2022; Li et al., 2023). By maintaining fMRI features as a persistent key–value memory, our design mitigates vanishing conditioning during long-form decoding and empirically yields stronger neural sensitivity and improved captioning performance compared to prefix-only baselines (Section 4).

### 3.4. Stage 1: Multi-Modal Contrastive Pre-Training

**Motivation** Subject-agnostic fMRI encoders often rely on location and parcellation priors to handle heterogeneous voxel layouts across subjects. In particular, neuroscience-informed attention can construct keys from voxel coordinates and ROI embeddings while using fMRI signals primarily in the value stream (Qiu et al., 2025). In our analysis, this can lead to a conditioning collapse failure mode where brain tokens (and downstream captions) change little under value perturbations such as voxel zeroing or permutation

*Table 1.* Cross-attention insertion ablations on Subject-8 (greedy decoding; temperature=0, top p=1). Metrics are computed with 5 human reference captions per image using the same evaluator. Indices denote the self-attention layer after which a cross-attention layer is inserted (as printed by the injector). All captioning metrics in this paper are reported ×100 for readability.

| Ablation | Placement (after SA layer indices) | #XAttn | BLEU-4 | ROUGE-L | METEOR | CIDEr |
|---|---|---|---|---|---|---|
| **Count** | Uniform (baseline) | 4 | 9.79 | 36.22 | 15.44 | 26.45 |
| | Uniform (baseline) | 7 | **10.53** | **36.32** | 15.46 | 25.86 |
| | Uniform (baseline) | 14 | 9.37 | 35.62 | **15.93** | 25.59 |
| **Placement @ #=7** | Early-half (1,3,5,7,9,11,13) | 7 | 9.18 | 35.42 | 15.79 | 25.72 |
| | Late-half (15,17,19,21,23,25,27) | 7 | 9.09 | 35.18 | 15.64 | 24.73 |
| **Placement @ #=4** | Early-biased (3,7,11,15) | 4 | 9.57 | 35.96 | 15.76 | **26.64** |
| | Late-biased (15,19,23,27) | 4 | 9.93 | 36.09 | 15.47 | 26.47 |

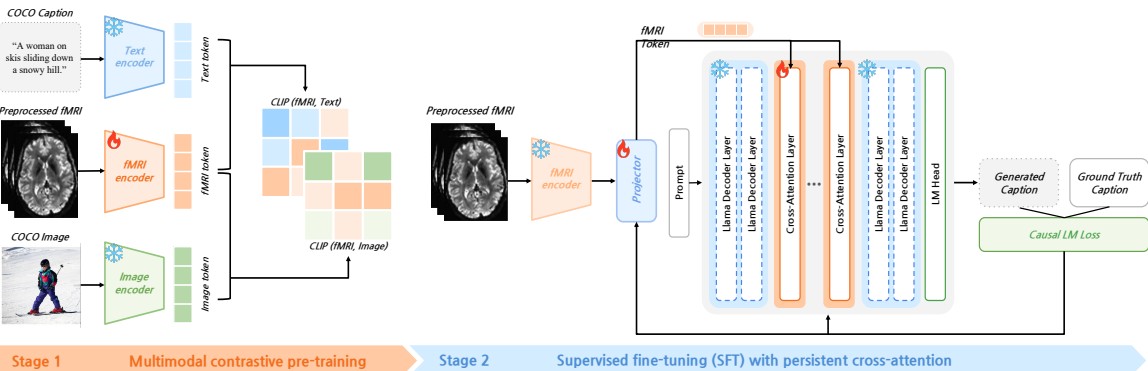

*Figure 3.* Training pipeline overview (Stages 1–2). Stage 1 aligns the fMRI encoder to a frozen CLIP space; Stage 2 trains the brain-LLM interface for caption generation with persistent brain cross-attention.

(Appendix B). Stage 1 therefore learns a stimulus-sensitive brain representation by explicitly aligning fMRI with paired image and text embeddings before training the generator.

We use Stage 1 to learn a semantically organized, subject-agnostic brain embedding space by aligning fMRI with both images and text using a CLIP-style symmetric InfoNCE objective. Contrastive language–image pretraining has shown that dual encoders can map modalities into a shared embedding space via in-batch negatives (Radford et al., 2021; Chen et al., 2020; He et al., 2020). Recent neuroimaging work has further explored mapping fMRI into CLIP-like spaces for decoding, retrieval, and reconstruction (Liu et al., 2023b; Peraza et al., 2025; Scotti et al., 2024).

Concretely, for each trial in NSD (Allen et al., 2022), we obtain an fMRI volume $x_{s,i}$ from subject $s$, the corresponding stimulus image $v_i$, and a natural-language caption $y_i$ describing $v_i$. We employ frozen CLIP-style encoders to compute $\ell_2$-normalized image and text embeddings $e_i^{\text{img}}, e_i^{\text{text}} \in \mathbb{R}^{d_c}$, and use the fMRI encoder (Qiu et al., 2025) with a small projection head to produce a brain embedding $e_{s,i}^{\text{fmri}} \in \mathbb{R}^{d_c}$. Thus, all three modalities reside in a common contrastive space of dimension $d_c$.

**Contrastive objective** We adopt a symmetric InfoNCE objective over in-batch samples, in the spirit of CLIP (Rad-

ford et al., 2021): For brain-image or brain-text alignment, let $\{(e_n^{z_1}, e_n^{z_2})\}_{n=1}^N$ denote a batch of normalised embeddings, where $N$ is the batch size. We define:

$$\mathcal{L}_{\text{con}} = -\frac{1}{N} \sum_{n=1}^{N} \left[ \log \frac{\exp\left(\langle e_n^{z_1}, e_n^{z_2}\rangle/\tau_c\right)}{\sum_{m=1}^{N} \exp\left(\langle e_n^{z_1}, e_m^{z_2}\rangle/\tau_c\right)} + \log \frac{\exp\left(\langle e_n^{z_2}, e_n^{z_1}\rangle/\tau_c\right)}{\sum_{m=1}^{N} \exp\left(\langle e_n^{z_2}, e_m^{z_1}\rangle/\tau_c\right)} \right] \quad (1)$$

where $z_1$ and $z_2$ denote the two modalities being aligned (e.g., brain–image or brain–text). For a batch of $N$ paired samples, $e_n^{z_1} \in \mathbb{R}^d$ and $e_n^{z_2} \in \mathbb{R}^d$ are the $\ell_2$-normalized embeddings produced by the corresponding projection heads, i.e., $\|e_n^{z_i}\|_2 = 1$. $\langle \cdot, \cdot \rangle$ denotes the dot product; because embeddings are normalized, $\langle e_n^{z_1}, e_m^{z_2} \rangle$ equals cosine similarity. $\tau_c$ is a (learned) contrastive temperature that scales the logits, following CLIP-style symmetric cross-entropy over in-batch negatives.

Our full multi-modal contrastive loss is

$$\mathcal{L}_{\text{contrastive}} = \mathcal{L}_{brain \leftrightarrow image} + \mathcal{L}_{brain \leftrightarrow text}. \quad (2)$$

In all Stage 1 experiments, we freeze the image and text encoders and update only the fMRI encoder and its projection head. We also pool trials from all subjects in the

same mini-batches, so that the contrastive objective encourages fMRI embeddings from different individuals to cluster by semantic content rather than subject identity. Stage 1 thus yields a brain encoder whose pooled embedding $e_{s,i}^{\text{fmri}}$ lives in a CLIP-aligned semantic space and whose token-level representation $B_{s,i}$ is later injected into the LLM via cross-attention as a persistent bank of brain tokens.

## 3.5. Stage 2: Supervised Fine-Tuning for fMRI-to-Text Generation

**Goal and training signal.** In Stage 2, we turn the pre-trained fMRI encoder and the cross-attention–augmented LLM into an end-to-end fMRI→text generator via supervised instruction fine-tuning, following the standard SFT paradigm used in instruction-tuned LMs and multimodal assistants (Ouyang et al., 2022; Liu et al., 2023a). Given an fMRI sample $x$ and its target caption $y = (y_1, \ldots, y_T)$, SFT trains the model to generate $y$ autoregressively under teacher forcing, conditioned on fMRI-derived brain tokens.

**Brain-token conditioning (interface recap)** The fMRI encoder produces brain tokens $\mathbf{B} = E_{\text{fmri}}(x) \in \mathbb{R}^{T_b \times d_e}$. We map them into the LLM hidden space with a lightweight projection and normalization:

$$\mathbf{Z} = \text{RMSNorm}(\mathbf{B}W_p) \in \mathbb{R}^{T_b \times d}, \qquad (3)$$

where $W_p$ is a trainable projection matrix and RM-SNorm (Zhang & Sennrich, 2019) denotes Root Mean Square Normalization. Text hidden states query $\mathbf{Z}$ through the interleaved brain cross-attention adapters defined in Section 3.3, which provide persistent access to neural tokens during generation (rather than one-shot prefix conditioning).

**Instruction format and response-only loss masking** Each training instance is serialized as an instruction–response example:

$$\texttt{[BOS] <INST>} \; p \; \texttt{</INST>} \; y \; \texttt{[EOS]}, \qquad (4)$$

where $p$ is a natural-language prompt (e.g., "Describe the image seen by the subject."), and $\texttt{[BOS]}/\texttt{[EOS]}$ denote the beginning-of-sentence and end-of-sentence tokens, respectively. Following common practice in supervised instruction fine-tuning, we compute the causal LM loss only over the response (caption) tokens, and mask prompt tokens from the objective (Ouyang et al., 2022; Liu et al., 2023a):

$$\mathcal{L}_{\text{SFT}} = - \sum_{t \in \mathcal{T}_{\text{resp}}} \log p_\theta(y_t \mid y_{<t}, p, \mathbf{Z}), \qquad (5)$$

where $\mathcal{T}_{\text{resp}}$ denotes the caption span.

**Parameter-efficient updates and stability** To preserve the pretrained linguistic competence of the backbone LLM,

we freeze the majority of LLM parameters and optimize only the brain-LLM interface:

$$\theta_{\text{train}} = \{W_p\} \cup \{W_Q^{(\ell)}, W_K^{(\ell)}, W_V^{(\ell)}, W_O^{(\ell)}, \alpha^{(\ell)}\}_{\ell \in \mathcal{L}}. \quad (6)$$

where $\mathcal{L}$ is the set of layers augmented with brain cross-attention (Section 3.3). We initialize the gating scalars $\alpha^{(\ell)}$ near zero so that, at initialization, the model behavior closely matches the original LLM and gradually learns to incorporate neural evidence; a stabilization strategy analogous to gated cross-attention in Flamingo-style adapters (Alayrac et al., 2022).

**No images are used in Stage 2** SFT optimizes the generator using only fMRI-derived brain tokens and text supervision; stimulus images and image features are neither accessed nor provided to the model during SFT.

**Objective** We freeze all LLM backbone parameters and update only the brain-LLM interface modules (the projector and cross-attention adapters, including gating parameters). This parameter-efficient design preserves the language prior while learning to condition generation on fMRI-derived tokens. BIT-LLM yields a neural-conditioned captioner that learns to follow task instructions while grounding generation in fMRI-derived tokens. In Stage 3, we further refine this model with reward optimization to directly target sequence-level caption quality and reference-based semantic fidelity.

## 3.6. Stage 3: Neuro-Semantic Reinforcement Fine-Tuning (RFT)

**Motivation** Stage 2 (SFT) aligns BIT-LLM with paired fMRI-caption supervision, but the resulting model can still over-rely on pretrained linguistic priors, producing fluent yet weakly grounded descriptions. To directly optimize sequence-level caption quality and reference-based semantic fidelity under the model's own generations, we introduce Neuro-Semantic Reinforcement Fine-Tuning (RFT). We adopt Group Relative Policy Optimization (GRPO) (Shao et al., 2024), a PPO-style post-training algorithm (Schulman et al., 2017) that enables efficient updates using group-relative comparisons and avoids an explicit critic. This choice is further supported by recent multimodal RFT results showing that reward-driven post-training can improve grounding (Liu et al., 2025). See Appendix E and Fig. 4 for the full GRPO pipeline.

**Trainable parameters and LoRA** In Stage 3, we freeze the fMRI encoder, the brain-to-LLM projector, the inserted cross-attention adapter weights, and the LLM backbone base weights. We update only LoRA parameters attached

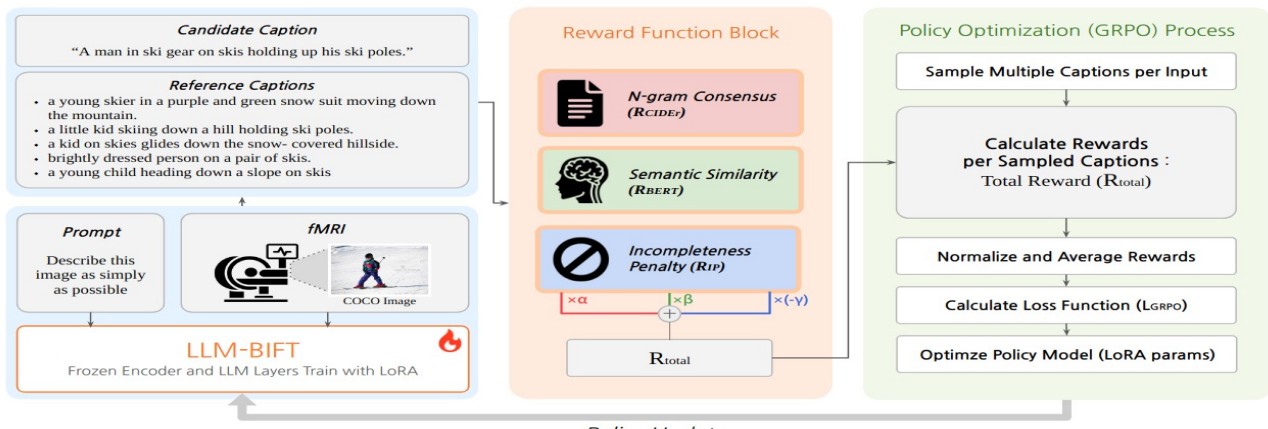

*Figure 4.* Overview of Stage-3 Neuro-Semantic RFT (GRPO). With the encoder and backbone LLM frozen, we sample multiple captions per fMRI input and compute a weighted reward combining n-gram consensus, semantic similarity, and an Incompleteness Penalty. Group-relative advantages across samples drive the policy (LoRA) update.

to selected LLM projections. Appendix E.11 reports a no-LoRA ablation where LoRA is disabled.

**Rollout decoding for GRPO (training) vs greedy decoding (evaluation)** Starting from the Stage 2 checkpoint, we maintain a trainable policy model $\pi_\theta$ and a frozen reference model $\pi_{\mathrm{ref}}$ (initialized from the same SFT weights) for KL regularization. During GRPO training, for each fMRI input we sample a group of $G$ candidate captions using stochastic decoding (temperature $\tau_{\mathrm{train}} > 0$ and nucleus sampling with $p_{\mathrm{train}} < 1$). In our experiments, we set $G = 4$ samples per prompt and perform 2 GRPO update iterations per rollout batch. At evaluation time, we report deterministic greedy decoding (temperature $= 0$, $p = 1$) for reproducibility.

**Reward design** For each sampled caption $\hat{y}$, we compute a scalar reward as a weighted sum of caption quality and semantic similarity:

$$R(\hat{y}) = \alpha R_{\mathrm{CIDEr}}(\hat{y}) + \beta R_{\mathrm{BERT}}(\hat{y}) - \gamma R_{\mathrm{IP}}(\hat{y}). \quad (7)$$

CIDEr encourages human-like descriptions via TF–IDF weighted $n$-gram consensus (Vedantam et al., 2015). BERTScore provides a semantic similarity signal that is less sensitive to paraphrasing than pure $n$-gram overlap (Zhang et al., 2020). The Incompleteness Penalty $R_{\mathrm{IP}}$ discourages premature termination or truncated outputs, such as captions ending with determiners (e.g., "a", "the") or missing sentence-final punctuation; Appendix E.10 quantifies its effect on incomplete endings. All GRPO hyperparameters are reported in Table 12, and reward-weight sensitivity is reported in Appendix E.3.

**Optimization objective** We optimize the GRPO objective (Shao et al., 2024), a PPO-style clipped surrogate with KL

regularization to a frozen reference policy (Schulman et al., 2017). Following GRPO, we sample $G$ candidates per input, compute rewards for all samples, and form group-relative advantages by normalizing rewards within each group (e.g., subtracting the group mean, optionally with variance normalization). Policy updates are applied to response tokens only (prompt tokens are masked as in Stage 2), using a clipped ratio with clip range $\epsilon$ and a KL penalty to the reference model with coefficient $\beta_{\mathrm{KL}} = 0.05$. In Stage 3 (default), all base weights (encoder, projector, cross-attention adapters, and LLM backbone) are frozen and GRPO updates are applied only to LoRA (Hu et al., 2022) parameters.

## 4. Experiments

Our experiments are organized around three questions: (i) can we improve held-out subject fMRI-to-caption decoding on NSD, (ii) do gains reflect end-to-end reliance on voxel values rather than language-prior shortcuts, and (iii) what is the contribution of each stage in our three-stage training pipeline?

### 4.1. Experimental Setup

**Dataset and held-out-subject protocol.** We evaluate BIT-LLM on the Natural Scenes Dataset (NSD) with COCO-style human captions, using five reference captions per image (Allen et al., 2022; Lin et al., 2014). Our main benchmark follows the held-out-subject protocol denoted S1–7→S8. Training uses only the training split from Subjects 1–7. Validation and checkpoint/model selection use a fixed disjoint 5% holdout from the S1-S7 training data. Subject 8 is completely excluded from training, validation, hyperparameter tuning, checkpoint selection, prompt tuning, and subject-specific adaptation. All final results are

evaluated only on the S8 test split. The final S8 evaluation uses the NSD shared-image test stimuli, which are never used for training or validation; therefore, this setting is both unseen-subject and unseen-image.

**fMRI Preprocessing and Voxel Selection.** We use NSD beta version 3 in the 1.8-mm volume-based preparation. Unless otherwise stated, voxel inputs are restricted to the subject-specific `nsdgeneral` visual-cortical mask. Other ROI maps, including `prf-visualrois`, `floc-places`, `floc-faces`, HCP-MMP1, and Kastner2015, are used only for ROI-level analyses and not as additional input selection criteria for the main captioning benchmark. Voxel-wise normalization statistics are computed from the training data only and then applied to validation and held-out test inputs. The NSD beta maps are not treated as a common cross-subject voxelwise template in our pipeline; therefore, transfer to S8 does not rely on direct voxelwise correspondence through a shared group-normalized space.

**Decoding and metrics.** Unless otherwise stated, we report greedy deterministic decoding (temperature $= 0$, top-$p = 1$) with a maximum of 128 new tokens. We evaluate captions against the 5 references using BLEU-$k$ (Papineni et al., 2002), ROUGE-L (Lin, 2004), METEOR (Banerjee & Lavie, 2005), CIDEr (Vedantam et al., 2015), and SPICE (Anderson et al., 2016), computed with the same evaluation scripts across all experiments.

**Reproducibility** Full training hyperparameters, prompts, and implementation details are reported in Appendix G.

**Release plan.** Upon publication, we plan to release the training and evaluation code on GitHub, and the trained model checkpoints and preprocessed data resources on Hugging Face, subject to dataset licensing and redistribution constraints.

**Note on concurrent protocols** Several recent brain-to-LLM/MLLM systems report within-subject or partially overlapping NSD settings; we include a protocol discussion in the Appendix and restrict numerical comparisons in the main paper to baselines that match the S1–7→S8 protocol.

### 4.2. Baselines

**Prior subject-agnostic baselines** We compare against MindBridge, UniBrain, and MindLLM under the same held-out-subject protocol.

**Our variants** Ours (SFT) denotes Stage 2. Ours (RFT) denotes Stage 2 followed by Stage 3 Neuro-Semantic Reinforcement Fine-Tuning (RFT) (Section 3.6). Unless oth-

*Table 2.* Main results on COCO captioning (NSD; S1–7→S8; held-out subject generalization). Baseline scores for Mind-Bridge/UniBrain/MindLLM are taken from MindLLM Table 3 under the same held-out protocol, whereas MindLLM Table 1 reports within-/seen-subject evaluation and is not directly comparable (Qiu et al., 2025).

| Metric ↑ | MindBridge | UniBrain | MindLLM | BIT-LLM |
|---|---|---|---|---|
| BLEU-1 | 39.84 | 41.90 | 44.60 | **56.12** |
| BLEU-2 | 19.55 | 19.67 | 24.04 | **35.72** |
| BLEU-3 | 9.29 | 8.89 | 12.79 | **22.58** |
| BLEU-4 | 5.24 | 4.33 | 7.52 | **15.10** |
| METEOR | 10.39 | 10.80 | 11.16 | **15.69** |
| ROUGE-L | 31.10 | 31.54 | 33.47 | **40.42** |
| CIDEr | 8.70 | 6.40 | 13.22 | **34.90** |
| SPICE | 2.67 | 2.39 | 3.82 | **7.13** |

*Table 3.* LOSO results on NSD for our cross-attention model (4 adapters; greedy decoding).

| Held-out subject (samples) | BLEU-4 | CIDEr | ROUGE-L |
|---|---|---|---|
| **S1** (3,000) | 12.21 | 35.39 | 38.65 |
| **S2** (3,000) | 10.52 | 28.39 | 37.41 |
| S3 (2,371) | 10.24 | 29.10 | 36.98 |
| S4 (2,188) | 10.42 | 29.23 | 36.93 |
| **S5** (3,000) | **12.90** | **36.77** | **39.84** |
| S6 (2,371) | 10.58 | 29.64 | 37.50 |
| **S7** (3,000) | 10.17 | 27.87 | 36.96 |
| S8 (2,188) | 9.57 | 26.64 | 35.96 |

erwise specified, **BIT-LLM** refers to the full model after Stage 3.

### 4.3. Results on NSD Held-Out Subject Captioning

Table 2 reports the main subject-heldout (S1–7→S8) results. BIT-LLM improves over prior subject-agnostic baselines on all reported captioning metrics under the same held-out-subject protocol.

In addition to the standard S1–7→S8 protocol, we report leave-one-subject-out (LOSO) performance for our final cross-attention model to characterize subject-wise generalization and variability across NSD (Table 3).

We mark subjects frequently used in prior NSD studies (S1/S2/S5/S7) and find substantial variability across held-out subjects, even under the same training-set size (3,000 samples each). In particular, S1 and S5 yield higher CIDEr, indicating that subject-wise factors can meaningfully affect cross-subject generalization (Xia et al., 2024; Mai & Zhang, 2023; Wang et al., 2024).

To probe task adaptability beyond COCO-style captioning, we additionally train task-specific SFT variants after Stage 1 on NSD-aligned COCO-QA and OK-VQA under the same S1–7→S8 held-out-subject split. These auxiliary results, reported in Appendix J, evaluate whether the Stage-1 brain rep-

resentation and cross-attention interface can be adapted to question-conditioned semantic readout, rather than whether the final captioning checkpoint performs zero-shot QA.

## 4.4. Ablations and Analysis

We evaluate whether the performance gains support our two core claims: **(A)** improvements reflect genuine neural grounding—dependence on voxel values rather than language priors; and **(B)** each training stage contributes meaningfully to the pipeline.

**Controlled Interface Ablation: Prefix vs. Cross-Attention** We first isolate the effect of the conditioning interface. We fix the Stage 1 encoder and LLM backbone, varying only the interface (Prefix vs. Cross-Attention) under identical Stage 2 SFT protocols. Panel A of Table 4 shows that the Prefix baseline performs well on Real inputs but can severely degrade under the Zero ablation. Because zeroing all voxels is an intentionally extreme perturbation and may introduce distribution shift (e.g., scale changes before normalization), we interpret Zero primarily as a stress test rather than a natural counterfactual. Importantly, both interfaces exhibit clear drops under Shuffle, which preserves the marginal voxel-value distribution within each sample while breaking value-location correspondence; this provides a more distribution-controlled diagnostic of voxel-value reliance versus prior-driven shortcuts. Under Zero, the prefix baseline nearly collapses, whereas cross-attention retains non-trivial captioning scores. Under Shuffle, both interfaces drop sharply, indicating that both rely on voxel-value structure rather than only marginal voxel statistics.

We further evaluate graded Gaussian perturbations to separate mild input noise from structure-destroying perturbations. As reported in Appendix K, performance is largely preserved under weak noise but degrades under stronger noise, and drops much more sharply under Zero and Shuffle.

**Capacity and conditioning-path control.** The two interfaces differ in both trainable capacity and conditioning path. Prefix conditioning is parameter-efficient and injects fMRI tokens only through the input sequence, whereas cross-attention uses additional trainable adapters but keeps fMRI tokens as an external key-value memory that can be queried at multiple depths. We therefore do not interpret the prefix-vs-cross-attention comparison as a pure parameter-matched comparison. To test whether the gain is explained by trainable interface capacity alone, we additionally train a parameter-matched prefix MLP under the same frozen encoder, frozen LLM backbone, NSD split, Stage 2 SFT recipe, greedy decoding, and evaluation pipeline. Even with a comparable number of trainable interface parameters, the prefix MLP underperforms cross-attention, suggesting

*Table 4.* Controlled interface comparisons on NSD S8. (A) Inference-time perturbations test whether generation depends on voxel values and value–location correspondence. (B) Real-input capacity control tests whether cross-attention gains are explained by trainable interface parameter count alone. All models use the same frozen Stage 1 encoder, frozen LLM backbone, Stage 2 SFT recipe, greedy decoding, and evaluation pipeline.

**(A) Inference-time perturbation**

| Interface | Cond. | BLEU-4 | CIDEr |
|---|---|---|---|
| Prefix | Real | 9.45 | 24.80 |
| | Zero | 0.00 | 0.02 |
| | Shuffle | 2.48 | 5.42 |
| XAttn | Real | 9.57 | 26.64 |
| | Zero | 4.36 | 5.98 |
| | Shuffle | 3.03 | 4.46 |

**(B) Real-input capacity control**

| Interface | Trainable params (M) | BLEU-4 | ROUGE-L | METEOR | CIDEr |
|---|---|---|---|---|---|
| Prefix (proj-only) | 3.15 | 9.45 | 34.99 | 15.12 | 24.80 |
| Prefix (parameter-matched MLP) | 100.67 | 7.69 | 33.59 | 14.58 | 23.98 |
| XAttn (Ours) | 103.82 | 9.57 | 35.96 | 15.76 | 26.64 |

that the improvement is not explained by interface capacity alone. We report attention-score compute proxies separately in Appendix F; these proxies are not full end-to-end FLOPs and are not used for the capacity-control claim in Table 4.

**Ablation A: Neural Reliance (Encoder Analysis)** Next, we verify if the encoder representation itself captures voxel values. We compare two Stage 2 models: one using our Stage 1 pretrained encoder, and another using the official baseline encoder checkpoint released by the authors (Qiu et al., 2025). We apply three inference-time conditions: Real, Zero (all 0), and Shuffle (intra-sample permutation). Table 5 shows that our Stage 1 encoder yields large performance gaps between Real and Zero/Shuffle inputs. This confirms end-to-end reliance on specific voxel values and their locations.

In contrast, the official baseline encoder is nearly invariant to perturbations, suggesting that its downstream captions are driven more by weakly input-dependent conditioning and language priors than by fine-grained voxel values. Visually, t-SNE analysis (Van der Maaten & Hinton, 2008) in Appendix C (Figure 6) further reveals that the encoder clusters brain samples by semantic category, validating the effectiveness of Stage 1 alignment.

**Additional Grounding Checks** (i) Retrieval: The Stage 1 encoder shows strong fMRI-to-image retrieval on Real inputs but fails on Zero/Shuffle (Appendix B), further validating voxel sensitivity. (ii) RFT Grounding: Even after Stage 3 RFT, the model maintains this sensitivity, with scores degrading significantly under perturbations (Table 15), indicating that RL fine-tuning does not override neural grounding. We also provide an auxiliary no-CIDEr RFT saliency

*Table 5.* **Ablation A (Encoder Reliance)** Our Stage 1 encoder shows distinct performance drops under perturbation, unlike the baseline which relies on priors.

| Encoder | Metric ↑ | Real | Zero | Shuffle |
|---|---|---|---|---|
| **Ours (Stage 1)** | BLEU-4 | 9.57 | 4.36 | 3.03 |
| | ROUGE-L | 35.96 | 28.49 | 27.53 |
| | METEOR | 15.76 | 11.73 | 9.59 |
| | CIDEr | 26.64 | 5.98 | 4.46 |
| **MindLLM-base** | BLEU-4 | 3.30 | 4.09 | 4.09 |
| | ROUGE-L | 30.51 | 31.19 | 31.19 |
| | METEOR | 11.04 | 11.31 | 11.31 |
| | CIDEr | 9.01 | 9.18 | 9.18 |

*Table 6.* **Ablation B (Stage-wise Contributions).** Each stage provides substantial gains.

| Metric ↑ | w/o Stage 1 | Stage 2 (SFT) | Stage 2+3 (RFT) |
|---|---|---|---|
| BLEU-1 | 34.16 | 46.46 | 56.12 |
| BLEU-2 | 11.85 | 26.14 | 35.72 |
| BLEU-3 | 2.52 | 15.52 | 22.58 |
| BLEU-4 | 0.38 | 9.57 | 15.10 |
| METEOR | 10.91 | 15.76 | 15.69 |
| ROUGE-L | 27.19 | 35.96 | 40.42 |
| CIDEr | 6.42 | 26.64 | 34.90 |

analysis in Appendix E.8, suggesting that the RFT-induced saliency increase is not explained solely by direct CIDEr optimization.

**Ablation B: Stage-wise Contributions**   Finally, Table 6 quantifies the value of each training stage. **Stage 1** is foundational; skipping it (w/o Stage 1) leads to poor generalization (CIDEr 6.42). **Stage 2** (SFT) aligns the interface, boosting CIDEr to 26.64. **Stage 3** (RFT) further refines the policy and improves the primary captioning metrics, increasing CIDEr from 26.64 to 34.90 and BLEU-4 from 9.57 to 15.10, while leaving METEOR essentially unchanged. As detailed in Appendix E.7, bootstrap analysis ($R = 10,000$) confirms that the improvements in CIDEr, BLEU-4, and ROUGE-L are statistically significant ($p < 0.05$).

## 5. Conclusion

We introduced **BIT-LLM**, a subject-agnostic framework for decoding fMRI into text that is designed to reduce the dominance of language priors in brain decoding. Instead of injecting neural features only once as a prefix, BIT-LLM keeps fMRI-derived tokens available throughout generation as a persistent memory of keys and values, which the decoder accesses through interleaved cross-attention. Comprehensive evaluations on the NSD benchmark with a held-out subject show that our three-stage pipeline, from contrastive alignment to reinforcement fine-tuning, achieves state-of-the-art captioning performance. Crucially, perturbation diagnostics show that these gains are accompanied by clear dependence

on voxel values, rather than being explained solely by linguistic shortcuts.

**Limitation**: First, reward design can be strengthened with neuro-visual verifiers to better detect grounding failures. Second, while we provide task-specific question-answering adaptation results in Appendix J, broader evaluation on zero-shot task transfer, multi-turn instruction following, and more diverse text-decoding tasks remains future work. Most importantly, cross-dataset generalization remains an important direction for future work. Transferring an NSD-trained decoder to heterogeneous datasets such as BOLD5000 (Chang et al., 2019) requires careful harmonization across scanner field strength, acquisition protocols, stimulus timing, preprocessing pipelines, and anatomical or functional alignment. We therefore leave rigorous cross-dataset evaluation to future work rather than treating preliminary transfer results as definitive evidence of universality.

Ultimately, scaling to broader subjects and modalities, alongside privacy-preserving protocols, will be essential to translate these advances into reliable general-purpose Neuro-AI interfaces.

## Acknowledgments

This work was supported by the National Research Foundation (NRF) grant funded by the Korea government (MSIT) (No. RS-2023-00218987; RS-2025-00562405; RS-2026-25518987), and in part by the Institute of Information & Communications Technology Planning & Evaluation (IITP) grant funded by the MSIT (No. RS-2026-25507282).

## Impact Statement

This paper advances methods for decoding visual perceptual content from non-invasive fMRI into natural language, with potential benefits for neuroscience, assistive communication, and the development of more reliable brain-conditioned AI systems. At the same time, brain-to-language modeling raises important ethical and societal considerations. Neural data can be privacy-sensitive, and progress in decoding methods may increase concerns about mental privacy, consent, and potential misuse (e.g., attempts at "mind reading," coercive inference, or surveillance). Our work uses research-grade fMRI acquired under controlled experimental conditions and does not enable covert or real-time extraction of private thoughts in everyday settings; nevertheless, the broader trajectory of neurotechnology warrants careful governance. We therefore emphasize (i) informed consent and clear data-use agreements, (ii) strong safeguards for storage and sharing of neuroimaging data (including de-identification and access controls), and (iii) evaluation practices that discourage overclaiming and reduce the risk of downstream misuse. We hope these results contribute

to scientific progress while motivating continued work on privacy-preserving learning, secure data stewardship, and responsible deployment standards for neural data and neurotechnology.

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

# A. Output example

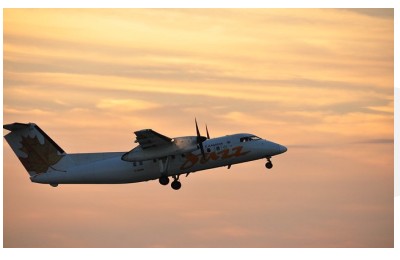 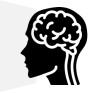

**Ground Truth Caption** (COCO ID-225829):
- an airplane with a maple leaf on it is flying through the air.
- a large propeller airplane flying through a cloudy sky.
- small white canadian commercial jet in flight at dusk.
- a medium engine plane flying across a sunset sky.
- a white plane with a leaf on the tale flying in the sky.

SFT (Stage 2): A plane is flying in front of a large mountain range.
**RFT (Stage 3)**: A large airplane flying in the sky.
Prefix (Llama 3.2 3b): A man sitting on a bench near a waterway.

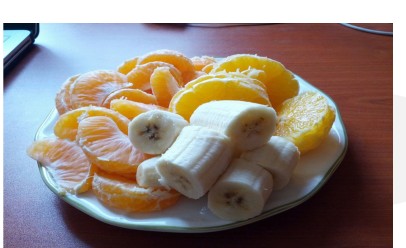 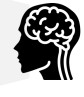

Ground Truth Caption (COCO ID-176085):
- a plate with several orange and banana slices on it.
- banana chunks go well with orange slices for a snack.
- the plate is full of banana and orange slices.
- the fresh fruit is on the plate ready to be eaten.
- a plate or orange and banana slices on a plate.

SFT (Stage 2): A table with two plates of food and two cups of coffee.
**RFT (Stage 3)**: A plate of food sitting on top of a table.
Prefix (Llama 3.2 3b): A stop sign on a pole with graffiti on it.

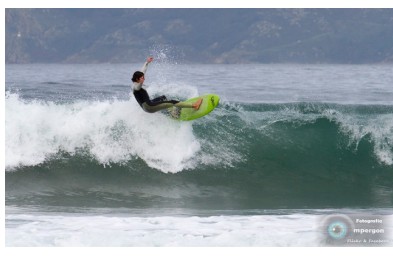

**Ground Truth Caption** (COCO ID-539879):
- the boy rides the ocean waves on a wake board.
- a surfer is in mid air while catching a wave.
- a person surfing on a medium sized wave.
- a surfer riding a wave in the ocean on a green surf board.
- a person riding a wave in the ocean on a surfboard.

SFT (Stage 2): A man riding a wave on top of a surfboard.
**RFT (Stage 3)**: A man riding a wave on top of a surfboard.
Prefix (Llama 3.2 3b): A train traveling down the tracks next to a building.

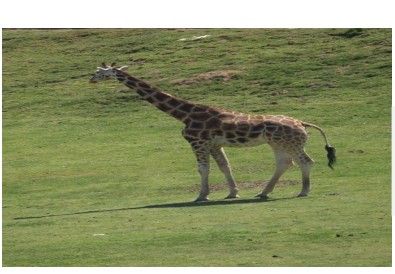 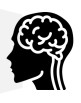

**Ground Truth Caption** (COCO ID-140590):
- a giraffe standing on top of a lush green field.
- a giraffe standing in the middle of a grass field.
- a giraffe in a field of grass next to a hill.
- the tall giraffe stands next to green, grassy hill.
- a giraffe standing and gazing on the ground.

SFT (Stage 2): A zebra standing in front of some trees and bushes.
**RFT (Stage 3)**: A giraffe standing on a field next to a tree.
Prefix (Llama 3.2 3b): A man is holding a large green umbrella.

*Figure 5.* Qualitative BIT-LLM results on NSD S8. Each panel shows the stimulus image, COCO references, and captions generated by BIT-LLM vs. a prefix baseline (Llama 3.2-3B). Prefix baseline is trained under the same controlled protocol as Table 4 (same Stage-1 encoder checkpoint and Stage-2 recipe); only the interface differs.

# B. Voxel-Value Sensitivity Analysis of fMRI Encoders

### B.1. Motivation

Strong downstream captioning performance does not necessarily imply that an fMRI encoder (or an end-to-end captioner) meaningfully utilizes voxel-level neural signals. In particular, an encoder can yield embeddings that are dominated by architectural components (e.g., positional/ROI embeddings, learned queries, or pooling) and thus become weakly sensitive to the actual voxel values. To explicitly probe value dependence, we adopt a real-versus-all-zero ablation as a randomization-style sanity check, analogous in spirit to input-invariance diagnostics used to reveal insensitivity to the intended input signal (Adebayo et al., 2018; Kindermans et al., 2017).

## B.2. Experimental Setup

**Encoders** We compare two frozen fMRI encoder checkpoints: (i) the MindLLM-base encoder checkpoint (`mindllm-base.ckpt`) released by (Qiu et al., 2025), and (ii) our encoder trained from scratch via Stage 1 multi-modal contrastive learning. While contrastive objectives (e.g., InfoNCE / NCE-style) are widely used for learning semantically aligned representations (Gutmann & Hyvärinen, 2010; Wang & Isola, 2020), they do not, by themselves, guarantee sensitivity to fine-grained input-value perturbations; we therefore validate value dependence explicitly.

**Data and evaluated sample counts.** We use preprocessed NSD beta volumes restricted to the shared-image set. Table 7 reports the number of shared-image trials used for each subject in this diagnostic. The maximum is 3,000 trials (1,000 shared images × up to 3 repeats); counts below 3,000 occur when a subject did not complete all sessions and therefore did not view the full shared set three times.

*Table 7.* Number of shared-image fMRI trials used in the voxel-ablation diagnostic for each NSD subject. The maximum is 3,000 trials (1,000 shared images × up to 3 repeats). Counts below 3,000 arise when a subject did not complete all sessions and thus did not view the full shared set three times.

| Subject | S1 | S2 | S3 | S4 | S5 | S6 | S7 | S8 |
|---|---|---|---|---|---|---|---|---|
| shared-image trials used | 3,000 | 3,000 | 2,371 | 2,188 | 3,000 | 2,371 | 3,000 | 2,188 |

## B.3. Ablation operators and protocol

Let $\mathbf{x}^{(k)} \in \mathbb{R}^V$ denote the cortical-mask voxel vector for the $k$-th sample of a subject. We consider the following inference-time ablations:

**(1) Real (`vox_ablate=none`)** Use $\mathbf{x}^{(k)}$ as-is.

**(2) Zero (`vox_ablate=zero`)** Set all voxel values to zero, i.e., $\tilde{\mathbf{x}}^{(k)} \leftarrow \mathbf{0}$. This removes voxel-value information while keeping the same positional/ROI embeddings and encoder configuration.

**(3) Intra-sample Shuffle (`vox_ablate=shuffle`)** We flatten the voxel tensor to 1D and (if provided) select only indices where `valid_vox_mask=True`. We then permute only those valid positions using `torch.randperm` (with a fixed seed for determinism) and write the permuted values back to the same valid indices; masked-out entries are left unchanged. Finally, we reshape back to the original tensor shape. This operation preserves the marginal distribution of valid voxel values within each sample, while breaking value-location correspondence.

## B.4. Representation similarity metric

We quantify voxel-value sensitivity using (i) cosine similarity and (ii) $\ell_2$ distance. We report subject-wise mean $\pm$ std across the evaluated samples. Intuitively, cosine similarity near 1 indicates near-invariance to voxel values, whereas similarity near 0 indicates strong value dependence. We report cosine similarity in tables for readability; $\ell_2$ distance yields the same qualitative conclusions.

## B.5. Results Across Subjects

Table 8 reports voxel-value sensitivity across subjects using cosine similarity. MindLLM-base (Qiu et al., 2025) yields near-invariant embeddings across all subjects (mean across subject means: $0.9889 \pm 0.0018$, range $[0.9866, 0.9925]$), suggesting that zeroing voxel values induces minimal change in the pooled embedding. In contrast, our Stage 1 pretrained encoder exhibits near-zero similarity (mean across subject means: $-0.0080 \pm 0.0345$, range $[-0.0527, 0.0368]$), consistent with strong dependence on voxel values. (Values slightly below zero are consistent with near-orthogonality under small stochastic variation.) Overall, Stage 1 pretraining produces representations that are substantially more value-dependent, whereas the MindLLM-base embeddings appear comparatively dominated by non-value components (e.g., priors from embeddings/queries/pooling).

*Table 8.* Voxel-value sensitivity across NSD subjects (S1–S8). We report cosine similarity between encoder embeddings from nonzero voxel inputs and all-zero voxel inputs. Near-1 similarity indicates invariance to voxel values; near-0 indicates strong value dependence.

| Subject | MindLLM-base mean ↑ | Ours (Stage 1) mean |
|---|---|---|
| 1 | 0.9891 | 0.0368 |
| 2 | 0.9889 | 0.0350 |
| 3 | 0.9925 | -0.0228 |
| 4 | 0.9887 | -0.0210 |
| 5 | 0.9874 | -0.0446 |
| 6 | 0.9898 | -0.0527 |
| 7 | 0.9879 | -0.0133 |
| 8 | 0.9866 | 0.0187 |
| Mean $\pm$ Std (across subjects) | $0.9889 \pm 0.0018$ | $-0.0080 \pm 0.0345$ |

## B.6. End-to-end Conditioning Sensitivity in Caption Generation

Encoder-level diagnostics are informative but indirect. To test whether an fMRI-conditioned captioner relies on fMRI inputs at inference time, we perform inference-time perturbation tests in the NSD subject-heldout setting (S1–7 train, S8 test) using an SFT-trained captioner. The main results are reported in Table 5 in the main text; here we specify the protocol and perturbation definitions for reproducibility.

**Controlled Stage 2 training protocol (encoder-only change).** We train two Stage 2 SFT captioners using an identical recipe (subject-heldout split, prompts/tokenization, interface architecture, optimizer settings, and number of updates), varying only the frozen fMRI encoder checkpoint: (i) MindLLM-base (`mindllm-base.ckpt`) (Qiu et al., 2025) and (ii) our Stage 1 pretrained encoder (`encoder_stage1.pt`). In all settings, the fMRI encoder and the LLM backbone are frozen during Stage 2 SFT; only the brain-LLM interface parameters are updated.

**Perturbations (inference-only).** Given a fixed trained captioner, we evaluate: (i) **Real**: unaltered fMRI; (ii) **Zero**: all voxel values set to zero; (iii) **Intra-sample Shuffle**: voxel values randomly permuted within each sample, preserving the marginal value distribution but breaking value-location correspondence (spatial patterns).

**Decoding, evaluation, and interpretation.** All perturbation conditions use **greedy decoding** (temperature=0, top-$p$=1) and are evaluated against the standard multi-reference captions (5 references per image), with scores multiplied by 100 for readability. Large drops under **Zero** indicate end-to-end reliance on voxel values, while additional drops under **Shuffle** indicate reliance on value-location correspondence beyond marginal statistics (Table 5).

## B.7. Encoder grounding via fMRI→image retrieval

Encoder-level perturbation diagnostics probe value dependence, but they do not directly test whether representations preserve stimulus semantics. We therefore evaluate the Stage 1 encoder on fMRI→image retrieval by ranking a fixed CLIP image-embedding gallery using cosine similarity.

*Table 9.* **Stage 1 encoder grounding via fMRI→image retrieval.** We retrieve the stimulus image by ranking a fixed CLIP image-embedding gallery using cosine similarity to the Stage 1 brain embedding. Real inputs yield meaningful retrieval (e.g., R@10=0.448; MedR=13), while voxel shuffling/zeroing collapses performance to near-chance (MedR≈500), indicating strong dependence on voxel values and their spatial correspondence (R@K: Recall at K, MedR: Median Rank, MRR: Mean Reciprocal Rank).

| Input | R@1 ↑ | R@5 ↑ | R@10 ↑ | MedR ↓ | MRR ↑ |
|---|---|---|---|---|---|
| Real | 0.1093 | 0.3170 | 0.4481 | 13 | 0.2166 |
| Shuffle | 0.0007 | 0.0053 | 0.0094 | 504 | 0.0074 |
| Zero | 0.0011 | 0.0052 | 0.0102 | 500 | 0.0076 |

## B.8. Discussion and Implications

This analysis supports three takeaways. First, strong downstream generation scores alone do not guarantee meaningful utilization of voxel-level signals; simple sanity checks can reveal representations that are weakly sensitive to the intended

input signal (Adebayo et al., 2018; Kindermans et al., 2017). Second, the real-versus-zero ablation provides a minimal encoder-level value-dependence test, and Stage 1 contrastive pretraining substantially increases encoder sensitivity to voxel values across all subjects (Table 8). Third, end-to-end perturbation tests should distinguish distributional removal (Zero) from pattern corruption (Intra-sample Shuffle). When paired with Table 5 in the main text, these diagnostics indicate that value-dependent encoder representations are necessary for the captioner to exhibit measurable end-to-end neural reliance through the brain-LLM interface. Together, these results motivate our Stage 1 design and support interfaces (e.g., cross-attention) that can repeatedly and selectively access value-dependent neural representations during decoding.

## C. LLM Model Comparison

**Effect of LLM Backbone Capacity**    MindLLM adopts Vicuna-7B as its backbone LLM (Qiu et al., 2025). To assess whether captioning quality is primarily driven by the intrinsic language capacity of the backbone (rather than the brain-to-language interface), we replace the Vicuna backbone with stronger off-the-shelf LLMs (Llama 3.1-8B and Llama 3.2-3B), while keeping the rest of the pipeline fixed: the same fMRI encoder checkpoint, the same interface training recipe (frozen encoder/LLM; interface-only updates), the same data split (S1–7 train, S8 test), and the same greedy decoding/evaluation protocol. Importantly, this is not an inference-time backbone substitution: each backbone is paired with its own independently trained checkpoint (trained from scratch under the same MindLLM recipe), and we evaluate using that checkpoint.

Table 10 shows that swapping the backbone alone does not consistently improve COCO-caption metrics.

This suggests that, in our setting, performance is primarily constrained by the informativeness of the brain tokens and the effectiveness of the brain-LLM interface, rather than by the standalone language modeling capacity of the backbone.

*Table 10.* Backbone LLM comparison under a controlled protocol. For each backbone, we train a separate model (i.e., a newly trained interface checkpoint per LLM) following the MindLLM pipeline, while keeping the rest fixed: the same fMRI encoder checkpoint, the same interface training recipe, the same NSD split (S1–7 train, S8 test), the same number of updates, and greedy decoding at evaluation. Thus, the comparison is not a plug-and-play backbone swap at inference time. Replacing Vicuna-7B with stronger LLMs does not yield consistent improvements, suggesting that the bottleneck lies in the brain-to-language interface rather than backbone capacity.

| Metric | Vicuna-7B (Llama 2-7B backbone) | Llama 3.1 8B | Llama 3.2 3B |
|---|---|---|---|
| BLEU-1 | 44.90 | 41.21 | 41.57 |
| BLEU-2 | 23.78 | 20.03 | 19.78 |
| BLEU-3 | 11.85 | 9.36 | 8.81 |
| BLEU-4 | 6.75 | 4.87 | 4.52 |
| ROUGE-L | 33.11 | 31.17 | 31.16 |
| CIDEr | 12.76 | 8.98 | 8.36 |
| METEOR | 11.02 | 10.23 | 10.03 |

## D. Model Architecture

**Overview**    BIT-LLM couples a subject-agnostic fMRI encoder with a frozen instruction-tuned LLM backbone. Given a preprocessed fMRI volume, the encoder produces a fixed-length set of brain tokens $\mathbf{B} \in \mathbb{R}^{N_b \times d_e}$ with $N_b=128$ and $d_e=1024$. A lightweight projector maps these tokens to the LLM hidden dimension $d=3072$: $\tilde{\mathbf{B}} = \text{RMSNorm}(\mathbf{B}\mathbf{W}_p)$ with $\mathbf{W}_p \in \mathbb{R}^{d_e \times d}$ (no bias). The LLM backbone is `meta-llama/Llama-3.2-3B-Instruct` (28 layers; hidden size 3072). Its base weights are frozen in Stages 2 and in our default Stage 3 setting; Stage 3 uses LoRA by default, and we additionally report a w/o-LoRA ablation that performs full-parameter GRPO updates to the backbone (Appendix E.11).

**Prefix vs. Cross-attention interfaces.**    We consider two ways of exposing $\tilde{\mathbf{B}}$ to the LLM. (*i*) **Prefix** prepends $\tilde{\mathbf{B}}$ as continuous input tokens at the first layer, analogous to prefix-tuning style conditioning with a frozen LM. (*ii*) **Cross-attn (ours)** treats $\tilde{\mathbf{B}}$ as an external key–value memory and injects trainable cross-attention adapters that allow intermediate text states to repeatedly query neural evidence throughout autoregressive decoding.

**Cross-attention adapter and stability.**    Let $\mathbf{H}_\ell \in \mathbb{R}^{T \times d}$ denote the text hidden states at LLM block $\ell$. At blocks where an adapter is inserted, we compute

$$\mathbf{H}'_\ell = \mathbf{H}_\ell + \alpha \cdot \text{CrossAttn}\Big(\text{RMSNorm}(\mathbf{H}_\ell),\ \tilde{\mathbf{B}}\Big),$$

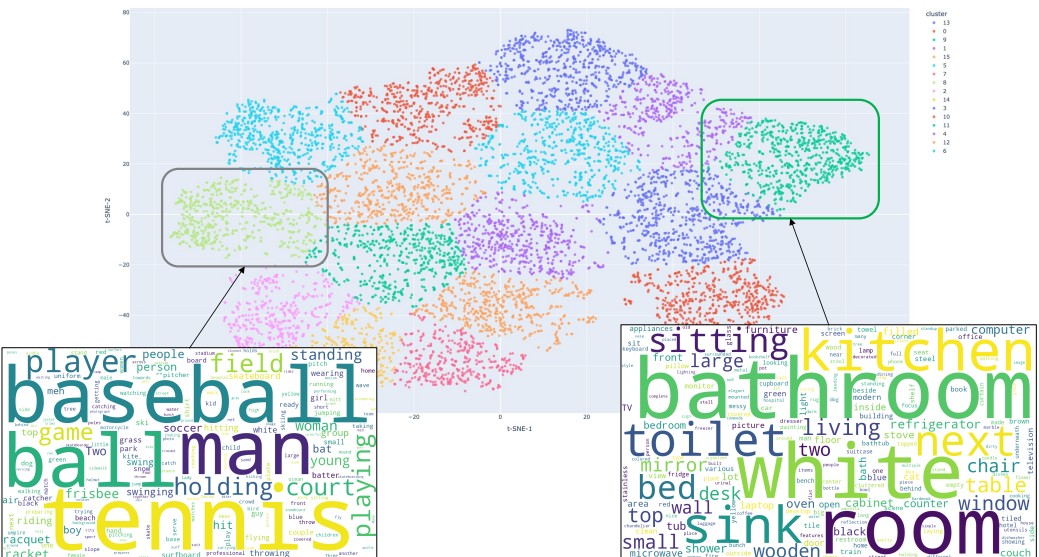

*Figure 6.* **t-SNE visualization ([Van der Maaten & Hinton, 2008](#)) of Stage 1 encoder representations.** We visualize the learned brain embeddings of test stimuli, colored by their COCO supercategories ([Lin et al., 2014](#)). The plot shows clear semantic clustering (e.g., distinct regions for furniture, animals, and vehicles), demonstrating that our contrastive pre-training successfully maps fMRI signals into a structured semantic space even before the LLM decoding stage.

where $\alpha$ is a learned scalar gate (`cross_alpha`, initialized to 0.08). Inside cross-attention, we apply a temperature $T{=}1.3$ to the attention logits to improve optimization stability.

**4+1 insertion schedule (4 adapters).** We insert **4** cross-attention adapters using a **4+1 schedule** over the 28-layer backbone: adapters are placed after transformer blocks $\{3, 7, 11, 15\}$ (0-indexed), yielding an approximately uniform stride over depth. Unless stated otherwise, this insertion pattern is fixed across experiments.

**Training-time parameterization.** Stage 2 (SFT) freezes the fMRI encoder and the LLM backbone, and updates only the brain-LLM interface (projector, cross-attention adapter weights, and `cross_alpha`). Stage 3 (GRPO) additionally trains LoRA parameters while keeping the backbone frozen.

## E. Neuro-Semantic RFT Details

### E.1. RFT overview and notation

We perform reinforcement fine-tuning (RFT) after Stage 2 (SFT) using Group Relative Policy Optimization (GRPO) ([Shao et al., 2024](#)), a PPO-style clipped policy-gradient method ([Schulman et al., 2017](#)) that uses within-group relative baselines and thus avoids training a separate critic. Our RFT pipeline follows the standard multi-sample post-training recipe used in recent multimodal RFT work (e.g., Visual-RFT) ([Liu et al., 2025](#)).

For each fMRI sample $x_i$ in a batch of size $B$, we compute brain tokens $\mathbf{Z}_i$ (Section 3.3) and sample a group of $G$ candidate captions $y_{i,1:G}$ from the rollout policy $\pi_{\theta_{\text{old}}}$ under a shared instruction prompt $p$. All log-probabilities below are conditioned on the same context $(p, \mathbf{Z}_i)$ and are computed on response tokens only (prompt tokens masked, as in Stage 2).

### E.2. Reward design: CIDEr + BERTScore + Incompleteness Penalty

Our central design choice is a composite reward that balances linguistic fluency and reference-based semantic fidelity:

$$R_{\text{total}}(x_i, y_{i,g}) = \alpha\, R_{\text{CIDEr}}(y_{i,g}, \mathcal{Y}_i) + \beta\, R_{\text{BERT}}(y_{i,g}, \mathcal{Y}_i) - \gamma\, R_{\text{IP}}(y_{i,g}) \tag{8}$$

where $\mathcal{Y}_i$ is the set of reference captions (five COCO captions). CIDEr measures consensus-based caption quality using TF–IDF weighted $n$-gram matching ([Vedantam et al., 2015](#)).

$R_{\mathrm{BERT}}$ uses BERTScore to measure semantic similarity via contextual embeddings and to better handle paraphrases beyond surface $n$-grams (Zhang et al., 2020).

$R_{\mathrm{IP}}$ gives a penalty when generated caption is incomplete (end with to, the, a, ..., etc). It helps to discourage reward hacking and stabilize training.

Unless otherwise stated, we use $\alpha = 3$, $\beta = 0.3$, and $\gamma = 0.4$ in all default Stage-3 experiments.

### E.3. Reward-Weight Sensitivity

To assess sensitivity to the Stage-3 reward weights, we perform one-factor-at-a-time ablations around the default setting $(\alpha, \beta, \gamma) = (3, 0.3, 0.4)$ for CIDEr, BERTScore, and the Incompleteness Penalty. All other Stage-3 settings are kept fixed. The results show that the gains are robust to reasonable reward weights, while removing useful reward terms or over-scaling them can degrade performance.

*Table 11.* One-factor-at-a-time Stage-3 reward-weight ablations around the default setting $(\alpha, \beta, \gamma) = (3, 0.3, 0.4)$.

| Varied term | Weight | BLEU-4 | ROUGE-L | CIDEr | METEOR |
|---|---|---|---|---|---|
| CIDEr weight $\alpha$ | 0 | 11.05 | 37.64 | 26.25 | 12.64 |
| | 1 | 14.86 | 40.42 | 34.89 | 15.76 |
| | 3 (default) | 15.10 | 40.42 | 34.90 | 15.69 |
| | 5 | 13.82 | 39.87 | 31.33 | 15.07 |
| BERTScore weight $\beta$ | 0 | 13.54 | 39.19 | 30.51 | 14.58 |
| | 0.1 | 13.64 | 39.30 | 32.26 | 14.86 |
| | 0.3 (default) | 15.10 | 40.42 | 34.90 | 15.69 |
| | 0.5 | 13.19 | 39.11 | 32.87 | 14.62 |
| IP weight $\gamma$ | 0 | 12.86 | 39.12 | 28.70 | 14.80 |
| | 0.4 (default) | 15.10 | 40.42 | 34.90 | 15.69 |
| | 0.8 | 13.95 | 39.99 | 34.30 | 15.18 |

Removing CIDEr reduces performance, but excessive CIDEr scaling is also harmful, suggesting that the Stage-3 gain is not simply due to aggressive CIDEr over-optimization. A moderate BERTScore weight provides a useful semantic auxiliary signal. For the Incompleteness Penalty, removing the penalty degrades both performance and output well-formedness, whereas an overly large penalty is also suboptimal.

### E.4. Reward shaping and outlier control (optional)

To discourage reward hacking and stabilize training, we optionally add small shaping terms: (i) a completion bonus for well-formed endings, (ii) an Incompleteness Penalty for truncated responses, and (iii) a genericness penalty that discourages overly generic captions dominated by frequent "safe" words. Additionally, we optionally gate out low-quality samples using a semantic threshold on BERTScore to avoid reinforcing degenerate generations. These options correspond to `completion_bonus`, `Incompleteness_penalty`, `generality_penalty`, and the semantic gating flags in our implementation.

### E.5. GRPO objective (overflow-safe form)

We use Group Relative Policy Optimization (GRPO), a PPO-style clipped policy-gradient method that computes advantages from within-group reward statistics and regularizes the policy toward a frozen reference model (Shao et al., 2024; Schulman et al., 2017).

**Notation.** For each input $x_i$ (with context $(p, \mathbf{Z}_i)$), we sample a group of $G$ responses $y_{i,1:G} \sim \pi_{\theta_{\mathrm{old}}}(\cdot \mid x_i)$ from the rollout (old) policy. Let $t$ index response tokens only (prompt tokens are masked, as in Stage 2). Define token-level

log-probabilities

$$\ell_\theta(i, g, t) := \log \pi_\theta\big(y_{i,g,t} \mid x_i, y_{i,g,<t}\big), \tag{9}$$

$$\ell_{\text{old}}(i, g, t) := \log \pi_{\theta_{\text{old}}}\big(y_{i,g,t} \mid x_i, y_{i,g,<t}\big), \tag{10}$$

$$\ell_{\text{ref}}(i, g, t) := \log \pi_{\text{ref}}\big(y_{i,g,t} \mid x_i, y_{i,g,<t}\big). \tag{11}$$

**Group-relative advantages.** Given scalar rewards $R_{i,g} = R_{\text{total}}(x_i, y_{i,g})$, we normalize within each group:

$$\mu_i := \frac{1}{G} \sum_{g=1}^{G} R_{i,g}, \qquad \sigma_i := \sqrt{\frac{1}{G} \sum_{g=1}^{G} (R_{i,g} - \mu_i)^2 + \varepsilon}, \tag{12}$$

$$A_{i,g} := \frac{R_{i,g} - \mu_i}{\sigma_i}. \tag{13}$$

**Clipped surrogate term (PPO-style).** We define a token-level importance ratio w.r.t. the rollout policy:

$$r_{i,g,t}(\theta) := \exp\big(\ell_\theta(i, g, t) - \ell_{\text{old}}(i, g, t)\big), \tag{14}$$

$$\bar{r}_{i,g,t}(\theta) := \text{clip}\big(r_{i,g,t}(\theta), \, 1 - \epsilon, \, 1 + \epsilon\big). \tag{15}$$

The clipped surrogate loss is

$$\mathcal{L}_{\text{clip}}(\theta) := -\mathbb{E}_{i,g,t}\Big[ \min\big(r_{i,g,t}(\theta)A_{i,g}, \, \bar{r}_{i,g,t}(\theta)A_{i,g}\big)\Big]. \tag{16}$$

**KL-to-reference regularization (direction and estimator)** We regularize the updated policy toward a frozen reference policy using the forward KL, $D_{\text{KL}}(\pi_\theta \,\|\, \pi_{\text{ref}})$ (i.e., expectation under the current policy). For a sampled response token $y_{i,g,t}$, define token-level log-probabilities

$$\ell_\theta(i, g, t) := \log \pi_\theta(y_{i,g,t} \mid x_i, y_{i,g,<t}), \tag{17}$$

$$\ell_{\text{ref}}(i, g, t) := \log \pi_{\text{ref}}(y_{i,g,t} \mid x_i, y_{i,g,<t}), \tag{18}$$

and the log-ratio (note the order)

$$\Delta_{i,g,t} := \ell_{\text{ref}}(i, g, t) - \ell_\theta(i, g, t) = \log \frac{\pi_{\text{ref}}(y_{i,g,t} \mid \cdot)}{\pi_\theta(y_{i,g,t} \mid \cdot)}. \tag{19}$$

We use the GRPO non-negative token-wise KL estimator

$$\widehat{D}_{\text{KL}}(i, g, t) := \exp(\Delta_{i,g,t}) - \Delta_{i,g,t} - 1 \; \geq 0. \tag{20}$$

With the expectation taken over samples from $\pi_\theta$, this estimator satisfies $\mathbb{E}_{y \sim \pi_\theta}\big[\widehat{D}_{\text{KL}}\big] = D_{\text{KL}}(\pi_\theta \,\|\, \pi_{\text{ref}})$.

**Final GRPO objective** Our optimization objective is

$$\mathcal{L}_{\text{GRPO}}(\theta) = \mathcal{L}_{\text{clip}}(\theta) + \beta_{\text{KL}} \, \mathbb{E}_{i,g,t}\big[\widehat{D}_{\text{KL}}(i, g, t)\big], \tag{21}$$

where $\epsilon$ is the clip range and $\beta_{\text{KL}}$ controls regularization strength. All expectations are computed over sampled groups and response tokens only.

### E.6. Training configuration and hyperparameters

Table 12 lists the key hyperparameters for reproducibility, including rollout sampling, GRPO optimization, and numerical stability settings.

*Table 12.* RFT hyperparameter categories. Concrete values used in the default run are reported in Appendix G.

| Category | Hyperparameters |
|---|---|
| Rollout sampling | group size $G$ (`samples_per_prompt`), temperature, top-$p$, max_new_tokens |
| Reward | $(\alpha, \beta, \gamma)$ for CIDEr/BERTScore/Incompleteness Penalty; genericness penalty; |
| GRPO / PPO-style | `grpo_iters`, clip range $\epsilon$, KL coefficient $\lambda_{KL}$, advantage $\varepsilon$ (std floor) |
| Optimization | AdamW (lr, weight decay), warmup steps/ratio, batch size $B$, gradient clipping norm |
| Precision / efficiency | bf16 or fp16; KV-cached decoding for grouped rollouts |

## E.7. Statistical Significance of RFT Improvements

To quantify the uncertainty of the improvements gained from Stage 3 (RFT) over Stage 2 (SFT), we compute 95% paired bootstrap confidence intervals with $R = 10,000$ replicates by resampling the $N = 2188$ test stimuli with replacement (Efron, 1986). Table 13 reports the mean improvement ($\Delta$) and the 95% confidence intervals. The results confirm that the gains in CIDEr, BLEU-4, and ROUGE-L are statistically significant (intervals do not include 0), whereas the change in METEOR is not significant.

*Table 13.* SFT→RFT improvements with paired bootstrap 95% confidence intervals on the test set (greedy decoding).

| Metric | SFT(A) | RFT(B) | $\Delta$(B−A) | 95% CI of $\Delta$ |
|---|---|---|---|---|
| CIDEr | 26.64 | 34.90 | 8.26 | [6.48, 10.47] |
| BLEU-4 | 9.57 | 15.10 | 5.53 | [4.70, 6.26] |
| ROUGE-L | 35.96 | 40.42 | 4.46 | [3.80, 5.12] |
| METEOR | 15.76 | 15.69 | -0.07 | [-0.46, 0.22] |

## E.8. CIDEr-specific contribution to saliency changes

Because the Stage-3 reward partly overlaps with standard captioning metrics, improvements in CIDEr should not by themselves be interpreted as evidence of stronger neural grounding. To examine whether the RFT-induced increase in neural reliance is explained solely by the CIDEr reward term, we trained an additional GRPO variant without the CIDEr reward and compared voxel-level saliency across SFT, no-CIDEr RFT, and full RFT. We compute saliency using Input×Gradient with respect to the fMRI input and report the mean relative change.

*Table 14.* Auxiliary saliency comparison for Stage-3 RFT. The no-CIDEr RFT model still increases voxel-level saliency relative to SFT, while full RFT provides an additional but not exclusive CIDEr-related contribution.

| Model comparison | Mean saliency change |
|---|---|
| SFT vs. no-CIDEr RFT | +19.7% |
| SFT vs. full RFT | +27.7% |
| CIDEr-specific additional effect | 28.9% |

The no-CIDEr RFT model still increases voxel-level saliency relative to SFT, suggesting that the saliency increase after RFT is not explained solely by direct CIDEr optimization. The full RFT model yields a larger increase, indicating that the CIDEr term remains useful but accounts for only part of the total saliency change. Across ROI atlases, the CIDEr-specific contribution remained modest (approximately 28–31%), and ROI-level saliency patterns were highly similar between no-CIDEr RFT and full RFT ($r = 0.990$–$0.999$). We therefore interpret GRPO as mainly amplifying reliance on broadly similar neural sources rather than reorganizing the spatial pattern of saliency. This analysis is auxiliary and should be interpreted together with the perturbation results in Table 15, not as standalone proof of neural grounding.

## E.9. Perturbation Sensitivity After Stage 3 RFT

To verify that GRPO-based RFT does not wash out neural reliance, we evaluate the final RFT model under inference-time voxel perturbations. All settings follow the same greedy decoding and COCO-caption evaluation protocol as in the main paper.

*Table 15.* **Stage 3 (RFT) captioning sensitivity under voxel perturbations**. After RFT, caption quality remains sensitive to voxel corruption: both Zero and Shuffle substantially degrade scores, indicating preserved voxel-value reliance.

| Input | BLEU-1 | BLEU-2 | BLEU-3 | BLEU-4 | ROUGE-L | CIDEr | METEOR |
|---|---|---|---|---|---|---|---|
| Real | 56.12 | 35.72 | 22.58 | 15.10 | 40.42 | 34.90 | 15.69 |
| Shuffle | 42.78 | 21.36 | 10.92 | 6.56 | 31.07 | 13.68 | 10.60 |
| Zero | 37.85 | 19.93 | 10.16 | 5.69 | 30.48 | 10.41 | 10.42 |

### E.10. Incompleteness Penalty Suppresses Incomplete-Ending Reward Hacking

During reward-based finetuning, we observed a failure mode where generated captions terminate with an incomplete function word (e.g., *"and"*, *"with"*), producing visibly unfinished sentences that can still obtain non-trivial automatic scores. To quantify this artifact, we define the **Incomplete-Ending Rate (IER)** as the fraction of generated captions whose last token belongs to the following set:

$$\mathcal{E} = \{\texttt{and, or, with, of, to, a, the, in, on,}$$
$$\texttt{for, at, by, from, as, an, into, over, under}\}.$$

We compute IER over all test-set generations ($N = 2188$) for RFT models trained with and without the Incompleteness Penalty. Unless otherwise stated, we lowercase captions and strip trailing punctuation before extracting the last whitespace-separated token.

*Table 16.* Incomplete-Ending Rate (IER). Adding the Incompleteness Penalty nearly eliminates incomplete endings ($50.96\% \rightarrow 0.37\%$), indicating strong suppression of a reward-hacking failure mode.

| Setting | #Captions ($N$) | #Hits | IER |
|---|---|---|---|
| RFT w/o IP | 2188 | 1115 | 0.5096 |
| RFT w/ IP | 2188 | 8 | 0.0037 |

**Metric trade-off.** For completeness, adding IP slightly decreases automatic captioning metrics in this run (e.g., CIDEr $36.7 \rightarrow 34.9$; BLEU-4 $15.13 \rightarrow 15.11$, reported ×100), but it enforces a basic well-formedness constraint and substantially improves output reliability.

**Qualitative examples**   Below we show representative generations that end with an incomplete token from $\mathcal{E}$.

**Without IP (frequent failures):**

- `A man sitting in front of a white` **and**

- `A stop sign sitting on top of a street with a white` **and**

- `A train sitting on the tracks with a white` **and**

- `A kitchen with a table sitting next to` **a**

- `A plate of food sitting on a table with` **a**

**Takeaway**   IP nearly eliminates incomplete-ending generations (Table 16), mitigating reward hacking that can otherwise exploit metric insensitivity to sentence completion.

### E.11. LoRA Ablation in Stage 3 Reward-Based Finetuning

**Protocol (controlled comparison)**   To isolate the effect of enabling LoRA during Stage 3 GRPO, we start from the same Stage 2 SFT checkpoint and keep the Stage 3 recipe fixed (data split, prompts, rollout group size, sampling settings, reward weights, KL regularization, optimizer, and training steps). In the default setting, the fMRI encoder, the brain-to-LLM projector, the cross-attention adapters, and the LLM base weights are frozen, and we update only LoRA parameters on the

specified LLM projections. In the w/o-LoRA ablation, we disable LoRA and unfreeze the backbone for full-parameter GRPO updates. All reported results use **greedy decoding** (temperature $= 0$, top-$p = 1$, max_new_tokens $= 128$) for reproducibility, while GRPO implementations use stochastic sampling (best-of-$G$, $G = 4$, $\tau_{\text{train}} = 0.7$, $p_{\text{train}} = 0.9$).

This design follows GRPO (Shao et al., 2024), a PPO-style clipped policy optimization objective (Schulman et al., 2017) commonly used in reward-based post-training for LLMs (Ouyang et al., 2022).

*Table 17.* LoRA ablation for Stage 3 reward-based finetuning (GRPO). All rows share the same Stage 2 SFT initialization and identical Stage 3 hyperparameters; we vary only whether the LLM backbone is adapted via full-parameter updates (w/o LoRA) or via LoRA residuals with frozen base weights (with LoRA). All results use greedy decoding and the same evaluation scripts.

| Setting (S8 test) | BLEU-4 | ROUGE-L | METEOR | CIDEr | Trainable Params | Notes |
|---|---|---|---|---|---|---|
| Stage 2 SFT (reference) | 9.57 | 35.96 | 15.76 | 26.64 | 103.8M | interface-only |
| Stage 3 GRPO (w/o LoRA) | 13.77 | 39.65 | 15.14 | 33.30 | 3.3B | full-backbone update |
| Stage 3 GRPO (with LoRA) | 15.10 | 40.42 | 15.69 | 34.90 | 197.7M | LoRA only |

**Discussion** Under an otherwise identical GRPO recipe, LoRA-based backbone adaptation yields the best captioning performance while training far fewer parameters than full-parameter updates. We therefore use LoRA as a default parameterization for Stage 3 to improve parameter efficiency and training stability. We do not claim that LoRA universally dominates full-parameter GRPO; rather, this controlled comparison suggests that restricting policy updates to a low-rank subspace can act as an effective regularizer in reward-based post-training, potentially reducing harmful drift from the pretrained policy.

**Interpretation and grounding check** Table 17 shows that enabling LoRA yields substantially higher captioning quality (e.g., CIDEr 34.90 vs 33.30) while training $\sim 17\times$ fewer parameters (197.7M vs 3.3B), suggesting that parameter-efficient adaptation provides sufficient capacity for GRPO improvements without requiring full backbone updates. However, higher text metrics can in principle arise from stronger language priors rather than improved neural dependence. For the default LoRA setting, we report this perturbation diagnostic in Table 15, where the Real–Zero and Real–Shuffle gaps remain large. We therefore use the LoRA setting for the final model and do not use the no-LoRA row for grounding claims. In our Stage 2 SFT model with the contrastively pretrained encoder, this diagnostic already exhibits large drops under Zero and Shuffle (Real $\gg$ Zero/Shuffle; e.g., CIDEr $26.64 \to 5.98/4.46$), indicating meaningful reliance on voxel values and value-location correspondence; Stage 3 gains should not come at the expense of these gaps.

## F. Interface parameter and compute trade-offs

We report a simple, transparent accounting of (i) trainable interface parameters and (ii) a coarse attention-score compute proxy to contextualize the controlled interface comparison. Throughout, we assume the fMRI encoder and LLM backbone are frozen, and only interface modules are trained. Let $d=3072$ be the LLM hidden size, $d_e=1024$ the encoder token dimension, $n=28$ the number of transformer layers, $T$ the text sequence length, $L$ the number of brain tokens (prefix length), and $N_{\text{cross}}$ the number of inserted cross-attention adapters.

The parameter-matched prefix MLP is used only as a capacity-control baseline in Table 4. It has the same prefix sequence length as the proj-only prefix baseline and therefore the same attention-score proxy, but its deeper projector introduces additional projection FLOPs. We therefore exclude it from the main-text compute comparison and do not treat it as an efficiency baseline.

**Trainable interface parameters**  The shared projector (linear + normalization) contains

$$P_{\text{proj}} = d_e d + d \quad (= 1024 \times 3072 + 3072 = 3{,}148{,}800). \tag{22}$$

Each cross-attention adapter in our implementation (GQA-style projections) adds

$$P_{\text{xattn}} = d^2 + 2(d_e d) + d^2 + d + 1 \quad (= 25{,}168{,}897) \tag{23}$$

corresponding to $W_q \in \mathbb{R}^{d \times d}$, $W_k, W_v \in \mathbb{R}^{d_e \times d}$, $W_o \in \mathbb{R}^{d \times d}$, a normalization vector ($d$), and a learned scalar gate. Thus, training four adapters yields

$$P_{\text{train}} = P_{\text{proj}} + N_{\text{cross}} P_{\text{xattn}} \quad (= 3.15\text{M} + 4 \times 25.17\text{M} = 103.82\text{M}). \tag{24}$$

By contrast, the prefix (proj-only) baseline trains only $P_{\text{proj}}$ (3.15M parameters).

**Attention-score compute proxy**  As a coarse proxy for attention compute and memory, we count the number of attention-score elements (pre-softmax) across layers. Prefix conditioning increases the effective sequence length from $T$ to $T+L$, so the proxy scales as

$$C_{\text{prefix}} = n(T+L)^2. \tag{25}$$

Cross-attention leaves the text self-attention length at $T$ and adds $N_{\text{cross}}$ cross-attention operations between text queries ($T$) and brain tokens ($L$):

$$C_{\text{xattn}} = nT^2 + N_{\text{cross}}(TL). \tag{26}$$

For the controlled interface setting ($T=128$, $L=128$, $N_{\text{cross}}=4$), we obtain $C_{\text{prefix}} = 28 \cdot 256^2 = 1{,}835{,}008$ and $C_{\text{xattn}} = 28 \cdot 128^2 + 4 \cdot (128 \cdot 128) = 524{,}288$, i.e., prefix is $\sim 3.5\times$ larger under this proxy. We stress that this proxy does not capture all FLOPs (e.g., projection/FFN costs), but it cleanly reflects the dominant quadratic sensitivity of prefix to the increased sequence length.

*Table 18.* Interface-level capacity–compute trade-offs with frozen encoder and LLM backbone. The compute proxy counts attention-score elements (pre-softmax) and highlights that prefix increases the quadratic self-attention term via sequence length, whereas cross-attention adds a linear $TL$ term at inserted layers.

| Interface (trainable modules) | Trainable params | Compute proxy ($C$) |
|---|---|---|
| Prefix (proj-only) | 3.15M | $28(T+L)^2$ |
| Cross-attn (4 adapters + proj) | 103.82M | $28T^2 + 4(TL)$ |

## G. Reproducibility and Implementation Details

### G.1. Prompt template (captioning)

We use a fixed two-message prompt template for all caption-generation experiments. A *system* message specifies the assistant role and output constraints, while a *user* message provides the task instruction. The exact prompts are:

**System:** You are a helpful agent that decodes the brain activity of a person looking at an image. Output exactly ONE short caption. Do NOT ask for images or more info. Do NOT mention fMRI/brain/limitations. No self-reference.
**User:** Describe the image as simply as possible.

We keep this template unchanged across methods and ablations to ensure that performance differences are attributable to the brain-LLM interface and training procedure rather than prompt variations.

### G.2. Backbone and interface

**LLM backbone**  We use `meta-llama/Llama-3.2-3B-Instruct` (Dubey et al., 2024) as the frozen backbone (28 layers; hidden size $d$=3072).

**Brain tokens and projection**  The fMRI encoder produces $N_b$=128 brain tokens of dimension 1024. We map them to the LLM hidden size using a linear projector `Linear(1024→3072, no bias)` followed by `RMSNorm`.

### G.3. Cross-attention interface details

For Cross-attn variants, we insert lightweight cross-attention adapters into the frozen backbone. We insert **4** adapters using a **4+1 schedule**: after transformer blocks $\{3, 7, 11, 15\}$ (0-indexed), yielding an approximately uniform stride over depth. Each adapter includes (i) a trainable scalar gate `cross_alpha` (init 0.08) and (ii) an attention temperature $T$=1.3 applied inside cross-attention.

**Notation**  Let $\mathbf{B} \in \mathbb{R}^{N_b \times d_e}$ denote encoder brain tokens ($N_b$=128, $d_e$=1024), and let $\tilde{\mathbf{B}} \in \mathbb{R}^{N_b \times d}$ be projected tokens in the LLM hidden size $d$=3072: $\tilde{\mathbf{B}} = \mathrm{RMSNorm}(\mathbf{B}\mathbf{W}_p)$ with $\mathbf{W}_p \in \mathbb{R}^{d_e \times d}$.

**Gated cross-attention update**  Given text hidden states $\mathbf{H} \in \mathbb{R}^{T \times d}$ at an insertion point, we compute

$$\mathbf{H}' = \mathbf{H} + \alpha \cdot \mathrm{Attn}\Big(\mathbf{Q} = \mathrm{RMSNorm}(\mathbf{H})\mathbf{W}_Q, \ \mathbf{K} = \tilde{\mathbf{B}}\mathbf{W}_K, \ \mathbf{V} = \tilde{\mathbf{B}}\mathbf{W}_V\Big) \mathbf{W}_O,$$

where $\alpha$ is a learned scalar gate (`cross_alpha`). Let $d_h$ denote the per-head dimension (i.e., $d_h = d/H$ for $H$ attention heads).

We apply RMSNorm to $\mathbf{H}$ before forming $\mathbf{Q}$, and use an attention-logit temperature $T$:

$$\mathrm{Attn}(\mathbf{Q}, \mathbf{K}, \mathbf{V}) = \mathrm{softmax}\left(\frac{\mathbf{Q}\mathbf{K}^\top}{T\sqrt{d_h}}\right) \mathbf{V}, \qquad T\text{=}1.3.$$

All adapter parameters $(\mathbf{W}_Q, \mathbf{W}_K, \mathbf{W}_V, \mathbf{W}_O, \alpha)$ are trained in Stage 2 while the LLM backbone weights remain frozen; in Stage 3 (default), these adapter weights are frozen and policy updates are parameterized via LoRA on selected backbone projections.

### G.4. Trainable modules by stage

*Table 19.* Trainable components by stage (default setting). The encoder and LLM base weights are frozen in all stages. Stage 2 trains the projector + brain cross-attention adapters; Stage 3 freezes the projector and cross-attention adapters and trains only LoRA on the frozen LLM backbone (default). Disabling LoRA in Stage 3 is used only as an ablation.

| Module | Stage-1 | Stage-2 (SFT) | Stage-3 (RFT/GRPO) |
|---|---|---|---|
| fMRI encoder | ✓ | ✗ | ✗ |
| Stage-1 CLIP projection head | ✓ | ✗ | ✗ |
| Brain-to-LLM projector $W_p$ | ✗ | ✓ | ✗ |
| Cross-attention adapters (+gate) | ✗ | ✓ | ✗ |
| LLM backbone weights $\theta_{\mathrm{LLM}}$ | ✗ | ✗ | ✗ |
| LoRA weights $\theta_{\mathrm{LoRA}}$ | ✗ | ✗ | ✓ |

## G.5. Stage 1: Encoder contrastive pretraining

**Goal.**    Stage 1 trains the fMRI encoder to produce stimulus-sensitive representations aligned with text and/or vision-language embeddings.

**Data and embeddings**    We use preprocessed NSD beta/voxel representations from the same pipeline as downstream training. We extract CLIP image/text embeddings using `openai/clip-vit-base-patch32` (OpenAI & Hugging Face, 2021; Radford et al., 2021). To avoid double dipping, we train using only the unique-image split used for training.

**Optimization and hyperparameters**    epochs=100; batch size=128; gradient accumulation=2; save_every=5.

## G.6. Stage 2: Supervised fine-tuning (SFT) for fMRI→caption

**Goal.**    Stage 2 trains the brain→LLM interface for caption generation conditioned on brain tokens.

**Frozen vs. trainable (default)**    The fMRI encoder is frozen after stage 1. In Stage 2 (SFT), we train the brain-LLM interface, consisting of the brain-to-LLM projector and the inserted brain cross-attention adapters (including the gating parameters, when used), while keeping the LLM backbone weights fixed. In Stage 3 (GRPO), our default updates only LoRA parameters on the frozen LLM backbone while keeping the encoder, projector, and cross-attention adapters frozen; the w/o-LoRA ablation disables LoRA and instead performs full-parameter GRPO updates to the backbone.

**Data split and validation.**    We follow the standard held-out subject protocol on NSD (Allen et al., 2022): train on subjects S1–S7 and evaluate on the held-out subject S8. We use `val_holdout`=0.05 with `val_seed`=42.

**Optimization**    epochs=10; batch size=32; gradient accumulation=2.

## G.7. Stage 3: Reinforcement fine-tuning (GRPO) with LoRA

**Goal and initialization.**    Stage 3 performs sequence-level policy optimization initialized from a Stage-2 checkpoint.

**Frozen vs. trainable**    The encoder is frozen (`--freeze_encoder`) and the LLM backbone is frozen (`--freeze_llm`); only LoRA parameters are trainable (`--use_lora`).

**LoRA configuration**    Targets: `q_proj,k_proj,v_proj,o_proj,gate_proj,up_proj,down_proj`; rank $r$=128, $\alpha$=128, dropout=0.05.

**GRPO hyperparameters**    epochs=30; batch size=32; samples_per_prompt=4; grpo_iters=2; learning rate=1e-6; KL coefficient $\beta = 0.05$.

**Decoding settings    Training (GRPO rollouts):** best-of-$G$ with stochastic decoding (temperature $\tau_{\text{train}}$=0.7, nucleus $p_{\text{train}}$=0.9).
**Evaluation:** greedy decoding (temperature= 0, top-$p$=1, max_new_tokens=128).

**Reward**    We use a composite reward:

$$R = 3 \cdot R_{\text{CIDEr}} + 0.3 \cdot R_{\text{BERT}} - 0.4 \cdot R_{\text{IP}},$$

where $R_{\text{CIDEr}}$ measures caption quality using TF-IDF weighted n-gram matching (Vedantam et al., 2015), $R_{\text{BERT}}$ measures semantic similarity (Zhang et al., 2020), and $R_{\text{IP}}$ gives a penalty when generated caption is incomplete.

## G.8. Decoding and evaluation (fixed across experiments)

**Deterministic decoding.**    All main results use greedy decoding: temperature= 0, top-$p$=1, max_new_tokens=128.

**Metrics**    We compute BLEU, ROUGE-L, METEOR, and CIDEr using standard COCO-caption evaluation. Since METEOR can be sensitive to Java/runtime versions, we log the evaluation environment.

## G.9. Compute

Stage-1 pretraining is run on 1×H200 GPU for approximately 4 days. Each Stage-2 SFT run is trained on a single H200 GPU. Stage-3 GRPO is also run on a single H200 GPU.

## G.10. Recommended controls for interface comparisons

For prefix vs. cross-attention comparisons, we recommend: (A) freeze the encoder and LLM for both methods and train only the interface modules; or (B) attach identical LoRA to the LLM in both variants and compare with/without cross-attention. This isolates conditioning-path effects from parameter-count differences.

*Table 20.* Reproducibility summary. "greedy" denotes temperature=0, top-$p$=1, max_new_tokens=128.

|  | Stage 1 (Pretrain) | Stage 2 (SFT) | Stage 3 (RFT/GRPO) |
|---|---|---|---|
| Trainable | Encoder | Proj + CrossAttn(+gate) | LoRA |
| Subjects | 1–7 | 1–7 | 1–7 |
| Held-out eval | – | subject 8 | subject 8 |
| Brain tokens | $128 \times 1024$ | $128 \times 1024$ | $128 \times 1024$ |
| Cross-attn | – | 4 (4+1) | 4 (4+1) |
| Gate / temp | – | train gate; $T$=1.3 | same |
| Decoding | – | greedy (0/1/128) | greedy (0/1/128) |

# H. Relation to concurrent brain-to-MLLM frameworks

Concurrent works connect fMRI representations to large pretrained generative backbones, often under the standard NSD within-subject protocol on the four fully-sampled subjects (subj01/02/05/07) with a shared-image test set (Xia et al., 2024; Mai & Zhang, 2023; Wang et al., 2024).

For example, UMBRAE aligns brain features to a frozen multimodal LLM and introduces a unified benchmark (Brain-Hub), reporting results on the standard NSD splits for these four subjects. MindBridge and Neuro-Vision-to-Language also study cross-subject decoding and brain-to-language interactions but are typically evaluated under different subject splits/tasks/metrics than our held-out-subject setting.

Because training/testing protocols differ (e.g., within-subject evaluation vs. train subjects 1–7 and test subject 8), direct numerical comparisons can be misleading; therefore, we discuss these works qualitatively and report quantitative comparisons only when protocols exactly match.

# I. Within-Subject Captioning Results

Although the main experiments focus on held-out-subject generalization, we additionally report within-subject captioning results for completeness. All models use the same tokenizer, decoding strategy, evaluation metrics, and caption evaluation pipeline.

*Table 21.* Within-subject BIT-LLM captioning results across NSD Subjects 1–7.

| Subject | BLEU-4 | ROUGE-L | CIDEr |
|---|---|---|---|
| S1 | 17.60 | 43.31 | 46.18 |
| S2 | 16.78 | 42.78 | 44.79 |
| S3 | 15.82 | 41.87 | 40.99 |
| S4 | 17.02 | 42.46 | 44.38 |
| S5 | 18.45 | 44.30 | 50.60 |
| S6 | 17.48 | 43.09 | 46.34 |
| S7 | 16.68 | 42.58 | 42.80 |
| Mean | 17.12 | 42.91 | 45.15 |

*Table 22.* Matched within-subject comparison using the same Llama 3.2 3B backbone and evaluation pipeline.

| Model | BLEU-1 | BLEU-2 | BLEU-3 | BLEU-4 | ROUGE-L | CIDEr | METEOR |
|---|---|---|---|---|---|---|---|
| MindLLM pipeline + Llama 3.2 3B | 57.02 | 37.42 | 24.35 | 16.25 | 42.37 | 43.94 | 16.79 |
| BIT-LLM | 58.85 | 39.47 | 25.70 | 17.12 | 42.91 | 45.15 | 17.36 |

These within-subject results are provided for completeness; the primary evaluation in this paper remains the subject-held-out S1–7→S8 setting.

## J. Task-Specific Question-Answering Adaptation

To evaluate whether the proposed brain representation and cross-attention interface can be adapted beyond COCO-style captioning, we train additional task-specific SFT variants on NSD-aligned COCO-QA and OK-VQA after Stage 1. These QA models share the same Stage-1 fMRI encoder design and held-out-subject protocol as the main captioning experiments, but they are not the final Stage-3 captioning checkpoint. Training uses only Subjects 1–7, and evaluation is performed on held-out Subject 8.

*Table 23.* Task-specific QA adaptation results under the S1–7→S8 held-out-subject protocol. These models are trained with QA-specific SFT after Stage 1 and are not the final captioning checkpoint.

| Task | MindBridge | UniBrain | MindLLM | BIT-LLM (QA-SFT) |
|---|---|---|---|---|
| COCO-QA | 35.88 | 24.95 | 38.96 | 48.52 |
| OK-VQA | 21.94 | 17.09 | 25.44 | 35.04 |

These results should be interpreted as auxiliary evidence of task adaptability, not as zero-shot QA performance of the captioning model or as a standalone test of neural grounding.

## K. Gaussian Noise Robustness

*Table 24.* Robustness to graded Gaussian noise on S8. Gaussian noise is added to normalized voxel inputs at inference time only.

| Input condition | BLEU-4 | ROUGE-L | CIDEr |
|---|---|---|---|
| Baseline | 9.57 | 35.96 | 26.64 |
| Gaussian $\sigma = 0.05$ | 9.61 | 36.00 | 26.33 |
| Gaussian $\sigma = 0.1$ | 9.69 | 36.08 | 26.49 |
| Gaussian $\sigma = 0.2$ | 9.63 | 36.03 | 26.06 |
| Gaussian $\sigma = 0.5$ | 9.19 | 35.43 | 24.56 |
| Gaussian $\sigma = 1.0$ | 7.97 | 33.76 | 18.86 |

We additionally evaluate graded Gaussian perturbations on normalized voxel inputs to characterize robustness under controlled input noise.

## L. Auxiliary ROI-Level Lesion Sanity Check

As an auxiliary model-level sanity check, we performed inference-time virtual lesioning within selected visual-cortical ROIs. This analysis is not intended as evidence for cortical causality. We zeroed voxel values within an ROI while keeping the trained model fixed, and compared the resulting degradation against size-matched random contiguous control masks sampled from the same subject-specific `nsdgeneral` mask.

*Table 25.* Auxiliary ROI-level virtual lesioning on NSD S8.

| Condition | Lesioned voxels | CIDEr | Drop |
|---|---|---|---|
| Baseline | 0 | 26.64 | – |
| Early visual (pRF) | 3491 (24.3%) | 22.10 | 17.04% |
| PPA | 808 (5.6%) | 20.04 | 24.77% |

Despite involving fewer voxels, the PPA lesion produced a larger CIDEr drop than the early-visual pRF mask. Among 100 size-matched random contiguous lesions, 48/100 were at least as disruptive as the early-visual lesion, whereas 0/100 matched the PPA lesion. We interpret this result only as an auxiliary diagnostic that the model is not solely sensitive to the number of perturbed voxels.

