# OpenReview forum: "BIT-LLM: Brain Instruction Tuned LLM with persistent Cross-Attention for fMRI-to-Text Decoding"
_ICML.cc/2026/Conference — ICML 2026 regular_

### Official Review · Reviewer_fg3j · 2026-03-11

**Soundness:** 3
**Presentation:** 3
**Significance:** 3
**Originality:** 3
**Overall Recommendation:** 5
**Confidence:** 5

**Summary:**

This paper proposes BIT-LLM for fMRI-to-text decoding, with innovations: 1) persistent neural access (i.e., use cross-attention instead of prefix-based modelling; 2) neuro-semantic reinforcement fine-tuning (RFT) with a specially designed reward function. The method shows strong performance on unseen subject generalization.

**Compliance With Llm Reviewing Policy:**

Affirmed.

**Final Justification:**

The proposed method makes nontrivial contributions to the domain and the experiments are comprehensive. However, it does not make a significant technical contribution to a machine learning conference. Therefore, I maintain a score of 5.

**Key Questions For Authors:**

The authors should specify the preprocessing procedures they use or indicate which prior work's preprocessing protocol they follow.

**Limitations:**

yes

**Strengths And Weaknesses:**

Strengths:
1. The method demonstrates strong performance, and the ablation studies comprehensively validate the sources of the performance improvements.
2. The findings are interesting. In particular, Table 5 provides insight into how much of the performance gain comes from the LLM’s language priors versus the semantic representations produced by the fMRI encoder.

Weakness:
1. The level of technical innovation appears limited. The proposed cross-attention mechanism and GRPO training do not introduce substantial methodological innovations beyond existing algorithms or methods.
2. I suggest that the authors include within-subject performance comparisons and evaluate text-decoding tasks in addition to captioning.

Minor:
1. Figure 4 appears blurry. Consider using vector graphics (e.g., PDF format) for the figures to improve clarity.
2. In the references, please ensure that fMRI is written in uppercase. In BibTeX, this can be enforced by enclosing it in curly braces.

---

> ### Author Rebuttal · Authors · 2026-03-31
>
> We sincerely thank the reviewer for the thoughtful and constructive feedback.
>
> **[W1]**
> We agree that our work does not primarily introduce a fundamentally new attention mechanism or optimization algorithm, and we will revise the paper to avoid overstating methodological novelty. Rather, our intended contribution is to show that, in the fMRI-to-text setting, treating brain activity as a first-class modality via cross-attention, rather than as a one-shot prefix, yields meaningful gains in decoding performance and grounding-oriented analyses. We will clarify this positioning more carefully in the revision.
>
> **[W2]**
> We agree that within-subject comparisons would be informative, and we appreciate the reviewer’s suggestion to broaden the evaluation beyond captioning.
>
> In response, we added within-subject captioning results across NSD Subjects 1–7 using the same tokenizer, evaluation metrics, and decoding settings. The subject-wise BIT-LLM results are:
>
> | Subject | BLEU-4 | ROUGE-L | CIDEr |
> |---|---:|---:|---:|
> | S1 | 17.60 | 43.31 | 46.18 |
> | S2 | 16.78 | 42.78 | 44.79 |
> | S3 | 15.82 | 41.87 | 40.99 |
> | S4 | 17.02 | 42.46 | 44.38 |
> | S5 | 18.45 | 44.30 | 50.60 |
> | S6 | 17.48 | 43.09 | 46.34 |
> | S7 | 16.68 | 42.58 | 42.80 |
> | Mean | 17.12 | 42.91 | 45.15 |
>
> For a more directly matched comparison, we also reproduced the MindLLM pipeline using the same LLaMA 3.2 3B backbone and evaluated it under the same within-subject setting. Under this matched comparison, BIT-LLM achieved higher mean performance on all reported metrics:
>
> | Model | BLEU-1 | BLEU-2 | BLEU-3 | BLEU-4 | ROUGE-L | CIDEr | METEOR |
> |---|---:|---:|---:|---:|---:|---:|---:|
> | MindLLM pipeline + LLaMA 3.2 3B | 57.02 | 37.42 | 24.35 | 16.25 | 42.37 | 43.94 | 16.79 |
> | BIT-LLM | 58.85 | 39.47 | 25.70 | 17.12 | 42.91 | 45.15 | 17.36 |
>
> We will add the full subject-wise comparison in the revision.
>
> At the same time, we would like to clarify that the primary objective of this paper is subject-held-out, subject-agnostic decoding, which is why the main submission focused on protocol-matched held-out evaluation rather than within-subject performance alone. This interpretation is also consistent with our encoder-level diagnostics: Table 8 shows that the official MindLLM-base encoder is nearly invariant to voxel zeroing, whereas our Stage-1 encoder changes substantially; Table 9 further supports the importance of stimulus-discriminative representations for grounded decoding.
>
> To address the reviewer’s second suggestion, we additionally evaluated the framework on question answering tasks aligned to NSD stimuli under the same subject-held-out protocol used in the main paper (train on Subjects 1–7 and evaluate on Subject 8):
>
> | Task | MindBridge | UniBrain | MindLLM | BIT-LLM |
> |---|---:|---:|---:|---:|
> | COCO-QA | 35.88 | 24.95 | 38.96 | 48.52 |
> | OK-VQA | 21.94 | 17.09 | 25.44 | 35.04 |
>
> These results suggest that the framework is not limited to caption generation, but can also support question-conditioned semantic readout from fMRI. For OK-VQA, we report the standard VQA soft accuracy rather than strict exact match, following the MindLLM evaluation protocol. We will add these results and the corresponding discussion to the revised manuscript.
>
>
> **[Minor]**
> We thank the reviewer for carefully noting these presentation issues. We will revise Figure 4 using a higher-quality vector format. We will also correct the reference formatting to ensure that {fMRI} is consistently capitalized throughout the bibliography.
>
> **[K1]**
> Thank you for this important comment. We agree that the provenance of the fMRI input should be described more explicitly.
>
> We used NSD beta version 3 within the ‘nsdgeneral’ visual mask, followed by additional voxel-wise standardization [1]. In the original NSD pipeline, beta version 3 was obtained from preprocessed fMRI data using temporal interpolation, spatial interpolation, voxel-wise HRF fitting, GLMdenoise, and ridge regression [2].
>
> We will revise the manuscript to describe this preprocessing protocol more explicitly. We will also clarify that ‘nsdgeneral’ was used for voxel selection, whereas the other ROI maps (HCP_MMP1, Kastner2015, ‘prf-visualrois’, ‘floc-places’, and ‘floc-faces’) were used only for ROI-level analyses.
>
> [1] Qiu, W., Huang, Z., Hu, H., Feng, A., Yan, Y., & Ying, R. (2025). MindLLM: A Subject-Agnostic and Versatile Model for fMRI-to-Text Decoding. arXiv preprint arXiv:2502.15786.
>
> [2] Allen, E.J., St-Yves, G., Wu, Y. et al. A massive 7T fMRI dataset to bridge cognitive neuroscience and artificial intelligence. Nat Neurosci 25, 116–126 (2022).

---

> > ### Author Rebuttal · Reviewer_fg3j · 2026-04-03
> >
> > I maintain my score.

---

### Official Review · Reviewer_tRcr · 2026-03-12

**Soundness:** 2
**Presentation:** 3
**Significance:** 3
**Originality:** 3
**Overall Recommendation:** 4
**Confidence:** 5

**Summary:**

The authors propose BIT-LLM, which replaces one-shot prefix conditioning with persistent cross-attention, and design a three-stage training pipeline consisting of contrastive learning pre-training, instruction fine-tuning, and reward fine-tuning. They achieve satisfactory results on the subject-agnostic fMRI-to-text task on NSD.

**Compliance With Llm Reviewing Policy:**

Affirmed.

**Final Justification:**

I will raise my score to 4

**Key Questions For Authors:**

(1) Please refer to the issues mentioned in the Weaknesses.

(2) I would like the authors to clarify the meaning of "S1–7→S8". Does it mean training on the S1–S7 training set, tuning parameters on the S1–S7 validation set, and testing on the S8 test set?

(3) I would like the authors to clarify the meaning of "Real/Zero/Shuffle inputs". In Table 4, are "Zero/Shuffle" trained on the corresponding inputs and tested on the respective models, or are they tested on a normally trained model? Additionally, I am curious why XAttn still achieves a certain level of score under Zero (compared to Prefix's near-zero score). Could the authors provide some output samples to illustrate under what circumstances the outputs receive a score of zero and under what circumstances they receive a certain score?

(4) Could the authors provide neuroscientific evidence to demonstrate that the model is genuinely better at capturing fMRI data, rather than merely leveraging a powerful LLM to produce clean captions from "noise" (e.g., voxel contribution heatmaps)?

**Limitations:**

yes

**Strengths And Weaknesses:**

### Sterngths

(1) The authors designed a novel persistent cross-attention instead of a prefix-style approach.

(2) The authors designed a three-stage training process (multimodal contrastive learning + instruction fine-tuning + reward fine-tuning) and experimentally verified the effectiveness of each stage.

(3) The authors conducted extensive experiments to support their claims, and the quantitative results demonstrate the superiority of the proposed method over previous works.

### Weakness

(1) The reward design in stage 3 closely aligns with the evaluation metrics. For instance, the reward includes CIDEr, and the results in Table 2 show that CIDEr exhibits the most significant improvement (>+160%, while other metrics show <+100% improvement compared to MindLLM). This raises a concern: how much of the gain from the proposed method stems from better neural grounding, versus being merely an improvement driven by optimization for the caption metric?

(2) The core task of this paper focuses on the reconstruction of COCO-style captions. However, such captions are highly structured text and represent a relatively simple task for modern LLMs. I believe the authors should consider more challenging tasks to comprehensively evaluate the model's ability to capture fMRI signals (e.g., grounding and image reconstruction tasks on the NSD dataset [1]).

(3) The comparison in Table 4 shows that XAttn has a significantly larger parameter count, yet the performance improvement on the Real metrics is not substantial. Although the authors claim that XAttn's attention operations are more efficient, the GFLOPs comparison does not support this assertion. I believe the authors should adopt a more equitable approach to determine whether the performance gains stem from the increased parameter count or from architectural/data flow changes.

(4) (minor weakness) The paper contains several formatting inconsistencies. For example: (a) The content in Table 4, Table 15, and the last line on page 18 exceeds the page width; (b) The heading formats appear inconsistent, with Section 4.1 in all caps while Section 3.3 uses lowercase...

---

[1] Xia et al., UMBRAE: Unified Multimodal Brain Decoding, ECCV 2024.

---

> ### Author Rebuttal · Authors · 2026-03-31
>
> We sincerely thank the reviewer for the feedback. We address each point below.
>
> **[W1]**
> We agree that the Stage-3 reward is partially aligned with the evaluation metrics, so the larger CIDEr gains in Table 2 should not by themselves be interpreted as proof of stronger neural grounding. To address this concern, we trained an additional GRPO model without the CIDEr reward and compared voxel-level saliency (Input×Grad) across SFT, no-CIDEr, and full RFT.
>
> | Model comparison | Mean saliency change |
> |---|---:|
> | SFT vs. no-CIDEr (RFT) | +19.7% |
> | SFT vs. RFT | +27.7% |
> | CIDEr-specific additional effect | 28.9% |
>
> These results indicate that CIDEr is useful, but the increase in neural-signal utilization under RFT is not explained by CIDEr alone. Higher saliency indicates greater sensitivity to fMRI perturbations, consistent with stronger neural reliance. Across ROI atlases, the CIDEr-specific contribution remained modest (28-31%), and ROI-level saliency patterns stayed highly similar ($r = 0.990$-$0.999$), suggesting that GRPO mainly amplifies reliance on broadly similar neural sources rather than reorganizing it. Consistently, Appendix E.7, Table 13 shows substantial post-RFT degradation under Zero and Shuffle, indicating preserved neural reliance after Stage 3.
>
>
> **[W2]**
> We agree that COCO-style caption reconstruction alone does not exhaustively evaluate semantic readout from fMRI. To address this concern, we additionally evaluated NSD-aligned VQA tasks under the same held-out-subject protocol (train on Subjects 1-7, evaluate on Subject 8).
>
> | Task | MindBridge | UniBrain | MindLLM | BIT-LLM |
> |---|---:|---:|---:|---:|
> | COCO-QA | 35.88 | 24.95 | 38.96 | **48.52** |
> | OK-VQA | 21.94 | 17.09 | 25.44 | **35.04** |
> These results suggest support for question-conditioned semantic readout from fMRI.
>
> **[W3]**
> To disentangle capacity from conditioning mechanism, we conducted a parameter-matched comparison under the same frozen-backbone Stage-2 protocol, with encoder, LLM backbone, NSD split, training, decoding, and evaluation fixed.
>
> | Interface                      | Params |       B4 |       R-L |    METEOR |     CIDEr |
> | ------------------------------ | ---------------: | -------: | --------: | --------: | --------: |
> | Prefix (proj-only)             |            3.15M |     9.45 |     34.99 |     15.12 |     24.80 |
> | Prefix (parameter-matched MLP) |          100.67M |     7.69 |     33.59 |     14.58 |     23.98 |
> | X-Attn |          103.82M | **9.57** | **35.96** | **15.76** | **26.64** |
>
> Even under this controlled setting, the parameter-matched prefix baseline underperformed cross-attention, suggesting that the gain cannot be explained by parameter count alone.
>
> **[W4]**
> We thank the reviewer for pointing this out. We have corrected formatting issues in the revision.
>
> **[K2]**
> “S1–7→S8” denotes a held-out-subject evaluation protocol on NSD. BIT-LLM was trained on subjects S1–S7 only, with validation within S1–S7, and test results were evaluated on subject S8.
>
> **[K3]**
> In Table 4, Real/Zero/Shuffle are not separately trained models; all three use the same trained model, with Zero/Shuffle applied only at inference time (Zero: all voxel values set to zero; Shuffle: intra-sample voxel permutation). Under Zero, XAttn typically collapses to a single generic caption (e.g., “A man in a suit and tie is standing in front of a building with trees behind it.”), so its nonzero score reflects occasional overlap with the references rather than stimulus-specific decoding. Under Shuffle, outputs remain fluent but are mostly incorrect. By contrast, Prefix under Zero often degenerates into malformed repetitive strings (e.g., “5 has 5 isabe 5 is 5 ...”), explaining its near-zero scores.
>
> **[K4]**
> To test whether the model relies on anatomically meaningful cortical signals rather than generic non-zero input, we performed inference-time virtual lesioning within visual-cortical ROIs and compared each ROI lesion against 100 size-matched random contiguous control masks within the same subject-specific nsdgeneral mask.
>
> | Condition             |   lesioned voxels | CIDEr |  Drop |
> | --------------------- | -----------: | ----: | -----: |
> | Baseline (unlesioned) | 0 | 26.64 |     -- |
> | Early visual (pRF)    | 3491 (24.3%) | 22.10 | 17.04% |
> | PPA                   |   808 (5.6%) | 20.04 | 24.77% |
>
> These results argue against a simple voxel-count explanation: despite containing many more voxels, the early-visual control lesion produced a smaller effect than PPA. Consistent with this, among 100 size-matched random contiguous lesions, 48/100 were at least as disruptive as the early-visual lesion, whereas 0/100 matched the PPA lesion. We also observed semantic drift under PPA lesioning; e.g., “A man riding a wave on top of a surfboard in the ocean.” changed to “A man riding skis down snow covered slope next to trees.” Together, these results support anatomically and functionally meaningful neural reliance during generation.

---

> > ### Author Rebuttal · Reviewer_tRcr · 2026-04-03
> >
> > The authors have addressed part of my concern, and I appreciate their clarification. Regarding the rebuttal, I still have the following two questions:
> >
> > (1) The rebuttal is still not sufficiently explicit for reproducibility. Please state clearly whether training uses only the training split of subjects S1–S7, validation uses only the validation split of subjects S1–S7, and testing uses only the test split of subject S8. In addition, please clarify whether this setting is truly an unseen-subject, unseen-image protocol. A related concern is that hyperparameter tuning is still performed on subjects S1–S7, i.e., on the same pool of subjects used for training. Please clarify how the authors ensure that model selection under this setup is predictive of performance on a truly new subject (S8), rather than benefiting from subject-specific tuning.
> >
> > (2) I remain unconvinced that the Real/Zero/Shuffle stress test is sufficient to characterize model stability, since both Zero and Shuffle completely disrupt the structure of the input fMRI. I would appreciate it if the authors could additionally evaluate the model under different levels of Gaussian noise on the input fMRI, which would provide a more controlled and informative stress test of robustness.

---

> > > ### Author Response · Authors · 2026-04-04
> > >
> > > We thank the reviewer for the helpful comments. We address the two points below and will revise the manuscript accordingly.
> > >
> > >
> > > **[R1]**  We thank the reviewer for requesting a more explicit description of the evaluation protocol. Our main experiments follow the standard NSD held-out-subject setting. Training uses only the training split of subjects S1–S7. Validation/model selection uses a disjoint fixed 5% holdout from the S1–S7 training data. All final results are evaluated only on the S8 test split, with S8 completely excluded from training, validation, hyperparameter tuning, and model selection.
> > >
> > >
> > > In our implementation, the final S8 evaluation uses the NSD 1,000 shared-image test set, and these images are never used for training or validation/model selection; under this split, the reported results are therefore both unseen-subject and unseen-image.
> > >
> > >
> > > Regarding model selection, we do not claim that validation on S1–S7 guarantees perfect prediction for every new subject; rather, it yields an unbiased held-out-subject evaluation because S8 remains completely untouched until final testing. All hyperparameters were selected only on S1–S7 and fixed before any S8 evaluation. We performed no S8-based model selection, prompt tuning, or subject-specific adaptation. Table 3 further reports LOSO results across subjects, suggesting that the protocol is not uniquely favorable to S8.
> > >
> > >
> > >
> > > Finally, to avoid any misunderstanding that transfer to S8 might be facilitated by a shared template representation, we clarify that the NSD beta maps used in our study are not represented in a common cross-subject template space (e.g., MNI). Specifically, we use NSD beta version 3 (b3) in the 1.8-mm volume-based preparation. The NSD Supplementary Methods state that the final functional outputs are volumetric fMRI time-series data in subject-native space [1]. Thus, the S1–S8 beta maps retain each subject’s native functional geometry and are not pre-aligned into a shared group-normalized space. Accordingly, transfer to S8 does not rely on direct cross-subject voxelwise correspondence through a shared normalized template space.
> > >
> > >
> > >
> > > [1] Allen, E.J., St-Yves, G., Wu, Y. et al. A massive 7T fMRI dataset to bridge cognitive neuroscience and artificial intelligence. Nat Neurosci 25, 116–126 (2022).
> > >
> > >
> > >
> > > **[R2]**  We thank the reviewer for this helpful suggestion. In addition to the Real/Zero/Shuffle stress test, we evaluated the model under Gaussian noise at multiple noise levels on the input fMRI, using the same caption evaluator across all settings. The results are shown below.
> > >
> > >
> > > | Input condition | BLEU-4 | ROUGE-L |  CIDEr |
> > > | --------------- | -----: | ------: | -----: |
> > > | Baseline        | 9.57 |  35.96 | 26.64 |
> > > | Gaussian 0.05   | 9.61 |  36.00 | 26.33 |
> > > | Gaussian 0.1    | 9.69 |  36.08 | 26.49 |
> > > | Gaussian 0.2    | 9.63 |  36.03 | 26.06 |
> > > | Gaussian 0.5    | 9.19 |  35.43 | 24.56 |
> > > | Gaussian 1.0    | 7.97 |  33.76 | 18.86 |
> > > | Shuffle         | 3.95 |  28.58 | 4.94 |
> > > | Zero            | 4.48 |  29.17 | 3.72 |
> > >
> > >
> > > These results provide a more controlled characterization of robustness. The normalized voxel inputs were approximately standardized (mean 0.02, std 0.98). Performance is largely preserved under weak Gaussian noise (std = 0.05, 0.1, 0.2), but degrades progressively under stronger perturbations (std = 0.5, 1.0). By contrast, the more extreme Zero and Shuffle conditions cause much larger drops. Overall, performance is largely preserved under small Gaussian perturbations, degrades under stronger noise, and drops much more sharply under the structure-destroying Zero and Shuffle conditions.
> > >
> > > We hope these clarifications and additional results address the reviewer’s concerns. We thank the reviewer again for the helpful feedback, which helped us improve the paper.

---

### Official Review · Reviewer_37zH · 2026-03-13

**Soundness:** 4
**Presentation:** 3
**Significance:** 3
**Originality:** 4
**Overall Recommendation:** 5
**Confidence:** 3

**Summary:**

This paper presents a 3-stage training framework for fMRI-to-text pipeline. The first stage applies contrastive learning to fine-tune a pretrained fMRI encoder to better suite the downstream task - image captioning; the second stage performs SFT training of the brain-LLM interface and the last stage uses RL fine-tuning to improve caption quality. The authors propose cross-attention mechanism for integration of fMRI tokens into the LLM architecture, inspired by other multimodal LLMs.
The approach shows significant gains in multiple metrics on a standart for this task NSD dataset, compared to a number of baselines. The authors additionaly provide mutliple ablations to isolate impact of each training stage and also "stress-tests" - zeroing or shuffling fMRI inputs - to demonstrate thah the model actually understands the fMRI signals.

**Compliance With Llm Reviewing Policy:**

Affirmed.

**Key Questions For Authors:**

1. Regarding Weakness 2, can the authors provide an ablation study on the choice of reward weights for CIDEr, BERTScore, and the Incompleteness Penalty?
2. Do the authors have any preliminary suggestions on how well would the method generalize to other datasets? For example, the mentioned BOLD5000.
3. Are the authors planning on releasing code or model weights, upon acceptance?

**Limitations:**

yes

**Strengths And Weaknesses:**

Strengths
1. Novel architecture.
Cross attention mechanism is applied to combine fMRI and text tokens. The approach is novel, compared to a previous idea of simply putting fMRI tokens as prefix. It is an intuitive way to condition a language model on a different modality and the approach shows score gains in comparison to a prefix baseline.
2. Well-motivated multi-stage training pipeline and ablation studies.
Each stage of the training pipeline is clear, technically sound and isolated impact of each stage is demonstrated by ablation studies, justifying the introduction of all components.
3. Strong empirical performance.
The approach achieves consistent state-of-the-art results across multiple metrics and is compared to a number of baselines. It suggests that the method is clearly a step forward for the fMRI-based image captioning task.
4. Stress tests.
"Stress-tests", conducted by the authors, such as zeroing fMRI inputs or shuffling voxels, demonstrate that the model meaningfuly relies on the signals. It can serve as a strong evidance that the model is sensitive to the change of the input and does not completely rely on prior-drive shortcuts.

Weaknesses
1. Evaluation is limited to a single dataset.
The experiments are conducted only on one dataset - NSD - and in a single experimental setup. Evaluation on a different dataset would provide even stronger evidance of superiority as well as generelizability of the approach. The authors acknowledge this weakness as a limitation.

2. Hyperparameters choice.
The authors provide configuration of hyperparameters but the impact and sensitivity of the model to their choice is not discussed. Partircularly, the ablation study on the reward weights would be a great contribution and boost confidence in the reported results.

3. Minor presentation issue.
Figure 4 seems to be of a little low visual quality and there is a spelling error - "Optimze" should be replaced with "Optimize", I believe.

4. No code release mentioned.
The authors do not explicitely mention whether they plan on releasing the codebase or model weights, upon acceptance?

---

> ### Author Rebuttal · Authors · 2026-03-31
>
> We sincerely thank the reviewer for the thoughtful feedback. Below, we address each comment in turn.
>
> **[K1, W2]**
>
> First, we note a minor text-only typo in Appendix G.7 of the submission. The text incorrectly listed the reward weights as $w_{\text{CIDEr}} = 3$, $w_{\text{BERT}} = 0.1$, and $w_{\text{IP}} = 0.3$, whereas the actual default hyperparameters used for all Stage-3 experiments were $w_{\text{CIDEr}} = 3$, $w_{\text{BERT}} = 0.3$, and $w_{\text{IP}} = 0.4$. We apologize for this oversight; it was a clerical error in the appendix text and does not affect any reported results or conclusions.
>
> To address your question, we performed one-factor-at-a-time ablations around these true defaults. The Stage-3 gains are robust to reasonable reward weights; extreme settings degrade performance.
>
> **CIDEr reward weight**: Removing CIDEr substantially reduces performance, but excessive scaling is also harmful. The best results lie in the moderate range (1-3).
>
> | $w_{\text{CIDEr}}$ | BLEU-4 | ROUGE-L | CIDEr | METEOR |
> |---|---:|---:|---:|---:|
> | 0 | 11.05 | 37.64 | 26.25 | 12.64 |
> | 1 | 14.86 | 40.42 | 34.89 | **15.76** |
> | 3 (default) | **15.10** | **40.42** | **34.90** | 15.69 |
> | 5 | 13.82 | 39.87 | 31.33 | 15.07 |
>
> These results suggest that the gains do not come from aggressively over-optimizing CIDEr; instead, a moderate CIDEr reward is beneficial, whereas excessive scaling is harmful.
>
> **BERTScore reward weight**: A moderate semantic reward performed best, confirming it as a useful auxiliary signal. Both smaller and larger values were less effective than the default setting.
>
> | $w_{\text{BERT}}$ | BLEU-4 | ROUGE-L | CIDEr | METEOR |
> |---|---:|---:|---:|---:|
> | 0 | 13.54 | 39.19 | 30.51 | 14.58 |
> | 0.1 | 13.64 | 39.30 | 32.26 | 14.86 |
> | 0.3 (default) | **15.10** | **40.42** | **34.90** | **15.69** |
> | 0.5 | 13.19 | 39.11 | 32.87 | 14.62 |
>
> **Incompleteness Penalty (IP)**: Removing IP reduced performance and reintroduced incomplete caption endings. Overly large IP was also suboptimal, reducing BLEU-4 and METEOR relative to the default $w_{\text{IP}} = 0.4$. This suggests that a moderate IP best suppresses reward hacking without over-penalizing valid generations.
>
> | $w_{\text{IP}}$ | BLEU-4 | ROUGE-L | CIDEr | METEOR |
> |---|---:|---:|---:|---:|
> | 0 | 12.86 | 39.12 | 28.70 | 14.80 |
> | 0.4 (default) | **15.10** | **40.42** | **34.90** | **15.69** |
> | 0.8 | 13.95 | 39.99 | 34.30 | 15.18 |
>
> In summary, these ablations show that the Stage-3 gains are robust to reasonable reward settings, with each reward term exhibiting the expected trade-off: removing a useful signal degrades performance, while over-scaling it is also harmful.
>
> **[K2, W1]**
> We agree that evaluation beyond NSD is important. In response, we conducted an additional zero-shot evaluation on BOLD5000, applying our NSD-trained model to the non-overlapping COCO subset of BOLD5000 **without any fine-tuning or parameter updates**. BIT-LLM outperformed a controlled MindLLM-style prefix baseline (with Llama 3.2 3B backbone) on average BLEU, CIDEr, and ROUGE-L across all four BOLD5000 subjects, with only a small deficit on METEOR.
>
> | Metric | MindLLM-style prefix (Avg. CSI1-CSI4) | BIT-LLM (Avg. CSI1-CSI4) |
> |---|---:|---:|
> | BLEU-1 | 39.16 | **44.25** |
> | BLEU-2 | 15.06 | **21.84** |
> | BLEU-3 | 3.52 | **11.56** |
> | BLEU-4 | 1.16 | **6.65** |
> | CIDEr | 4.07 | **9.52** |
> | ROUGE-L | 29.11 | **31.58** |
> | METEOR | **11.07** | 10.89 |
>
> The same overall trend was also observed at the subject-wise level across CSI1-CSI4, especially for BLEU-4 and CIDEr. While absolute scores are lower than in-dataset evaluations due to substantial domain shifts (e.g., 7T vs. 3T field strength, signal-to-noise ratio, and stimulus timing protocols), BIT-LLM consistently demonstrates stronger transfer performance. Our current transfer pipeline relies primarily on anatomical alignment; prior work [1] suggests that explicit functional alignment is more effective for fine-grained decoding. We therefore view these zero-shot results as preliminary but encouraging evidence of cross-dataset generalization, while recognizing that explicit functional alignment is a promising direction for further improving robustness and transfer performance.
>
> [1] Ferrante et al. Through their eyes: Multi-subject brain decoding with simple alignment techniques. Imaging Neuroscience (2024).
>
> **[K3, W4]**
> Yes. Upon acceptance, we plan to release the training and evaluation code on GitHub. We also plan to make the trained model checkpoints and the preprocessed data resources used in this work available on Hugging Face, including the preprocessed data resources used in this work.
>
> **[W3]**
> Thank you for pointing this out. We will correct the typo and improve Figure 4 in the revision.

---

### Decision · Program_Chairs · 2026-04-30

**Decision:**

Accept (regular)

**Comment:**

The paper received positive reviews, with two reviewers (37zH, fg3j) recommending accept and one reviewer (tRcr) recommending weak accept. The AC has examined the reviews, author responses, and the discussions. After rebuttal, the reviewers acknowledge their concerns have been addressed. The AC concurs and decides to accept the work, with belief that while the work is useful, the application domain of the proposed method is relatively narrow. The authors are encouraged to incorporate the promised clarifications and additional results in the final version.